# Investigating the sources of variable impact of pathogenic variants in monogenic metabolic conditions

Angela Wei [1,2,3,4], Richard Border [5,6], Boyang Fu [5], Sinéad Cullina[7,8], Nadav Brandes [9,10,11], Seon-Kyeong Jang[6], Sriram Sankararaman [3,4,5], Eimear E. Kenny[7,8,12,13], Miriam S. Udler [14,15], Vasilis Ntranos [9,10,11], Noah Zaitlen[3,4,6,16] & Valerie A. Arboleda [1,2,3,4,16] ✉

Over three percent of people carry a dominant pathogenic variant, yet only a fraction of carriers develop disease. Disease phenotypes from carriers of variants in the same gene range from mild to severe. Here, we investigate underlying mechanisms for this heterogeneity: variable variant effect sizes, carrier polygenic backgrounds, and modulation of carrier effect by genetic background (marginal epistasis). We leveraged exomes and clinical phenotypes from the UK Biobank and the Mt. Sinai Bio*Me* Biobank to identify carriers of pathogenic variants affecting cardiometabolic traits. We employed recently developed methods to study these cohorts, observing strong statistical support and clinical translational potential for all three mechanisms of variable carrier penetrance and disease severity. For example, scores from our recent model of variant pathogenicity were tightly correlated with phenotype amongst clinical variant carriers, they predicted effects of variants of unknown significance, and they distinguished gain- from loss-of-function variants. We also found that polygenic scores modify phenotypes amongst pathogenic carriers and that genetic background additionally alters the effects of pathogenic variants through interactions.

With the rapidly increasing use of exome sequencing in clinical practice, and with over three percent of the population carrying a pathogenic variant in genes associated with autosomal dominant disease[1–3], predicting which carriers will develop disease and how that severe the disease will manifest are central questions for the practice of genomic medicine[4,5] (Fig. 1A). Addressing the full spectrum of clinical genotypes associated with liability to diseases would improve preventative and targeted approaches prior to disease onset. However, the causes that

[1]Interdepartmental Bioinformatics Program, UCLA, Los Angeles, CA, USA. [2]Department of Pathology and Laboratory Medicine, David Geffen School of Medicine, UCLA, Los Angeles, CA, USA. [3]Department of Computational Medicine, David Geffen School of Medicine, UCLA, Los Angeles, CA, USA. [4]Department of Human Genetics, David Geffen School of Medicine, UCLA, Los Angeles, CA, USA. [5]Department of Computer Science, UCLA, Los Angeles, CA, USA. [6]Department of Neurology, David Geffen School of Medicine, UCLA, Los Angeles, CA, USA. [7]Institute for Genomic Health, Icahn School of Medicine at Mount Sinai, New York, NY, USA. [8]Department of Genetics and Genomic Sciences, Icahn School of Medicine at Mount Sinai, New York, NY, USA. [9]Department of Epidemiology & Biostatistics, UCSF, San Francisco, CA, USA. [10]Department of Bioengineering & Therapeutic Sciences (HIVE), UCSF, San Francisco, CA, USA. [11]Bakar Computational Health Sciences Institute, UCSF, San Francisco, CA, USA. [12]Division of Genomic Medicine, Department of Medicine, Icahn School of Medicine at Mount Sinai, New York, NY, USA. [13]Center for Translational Genomics, Icahn School of Medicine at Mount Sinai, New York, NY, USA. [14]Center for Genomic Medicine, Massachusetts General Hospital, Boston, MA, USA. [15]The Broad Institute, Boston, MA, USA. [16]These authors jointly supervised this work: Noah Zaitlen, Valerie A. Arboleda. ✉e-mail: vaa2001@g.ucla.edu

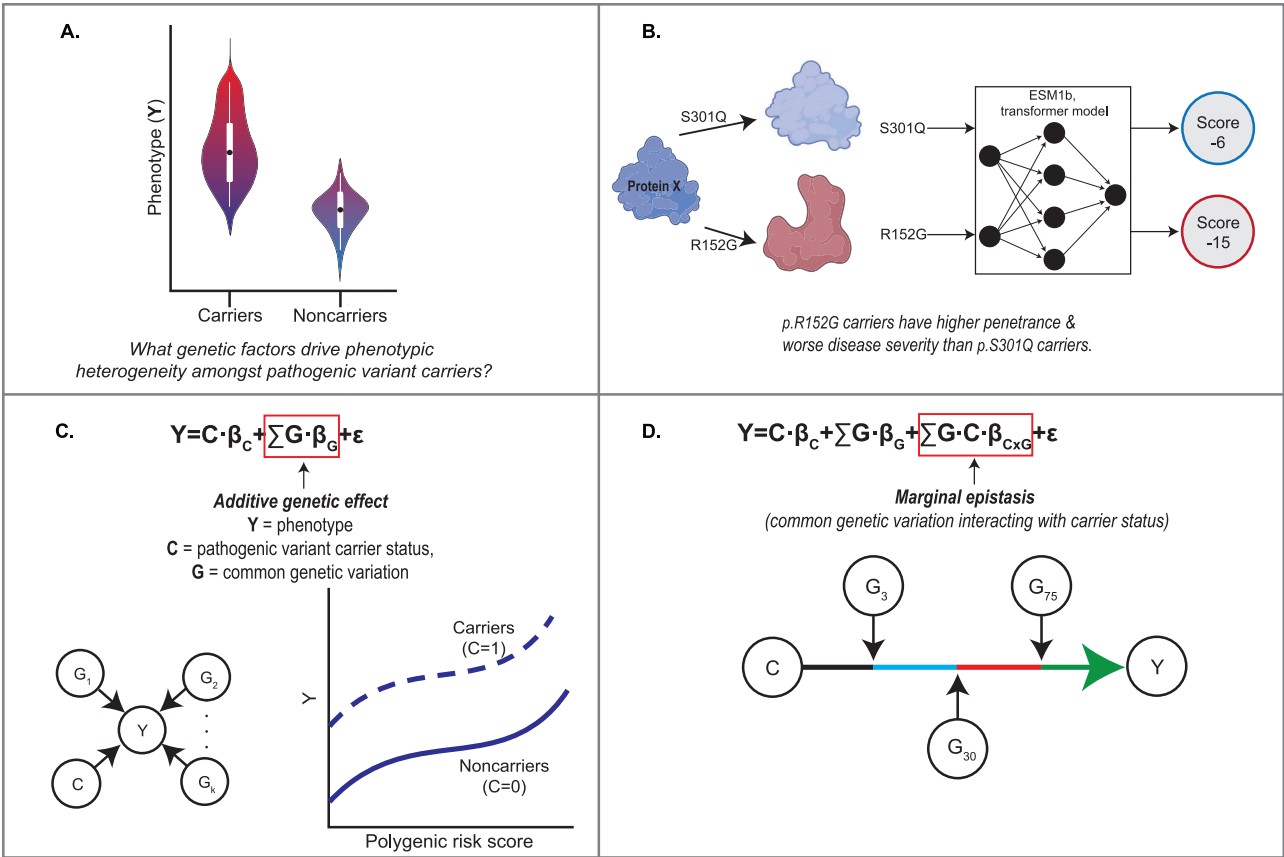

**Fig. 1 | Outline of study. A** Phenotypic heterogeneity exits within carriers and noncarriers of pathogenic variants; individuals will range from mild to severe diseases, as shown in this toy figure. This study applies novel bioinformatic methods to understand the genetic factors that affect carrier phenotype at biobank-scale. **B** We apply ESM1b, a protein language model, to predict the variable effect sizes of monogenic missense variants. **C** We utilize polygenic risk scores (PRS) to determine if pathogenic variant carrier phenotype is modified by additive genetic effects and identify the distribution of PRS where noncarriers have greater more severe phenotypes than carriers. Here, **Y** is the individual's phenotype, **C** indicates pathogenic variant carrier status and its effect size $\beta_C$, and **G** is common genetic variation with its effect size $\beta_G$. **D** We employ a novel method, FAME, to estimate the contribution of marginal epistasis ($\sum G \cdot C \cdot \beta_{CxG}$, interaction term) between carrier status and polygenic background to phenotypic variation. Created in BioRender. Arboleda (2025) https://BioRender.com/hmwost8.

affect penetrance and severity are largely unknown, making it difficult to determine which patients will require clinical interventions and what degree of intervention will be needed[5,6]. In this study, we applied recently developed computational methods to biobank-level data to study three theorized sources of this heterogeneity in the context of clinical metabolic traits: differing pathogenic variant effects within a gene, variable polygenic background amongst carriers, and marginal epistasis—the combined pairwise interaction effects between carrier status and all other SNPs while controlling for linear, additive effects (Fig. 1).

Mounting evidence suggests that each of these factors contribute to incomplete penetrance and variable disease severity. For example, loss-of-function (LOF) variants within the *MC4R* gene cause monogenic obesity; however, other missense variants in the same gene that are gain-of-function (GOF) are associated with protection against obesity[7]. Recently, Goodrich et al.[4], Fahed et al.[8] and Huerta-Chagoya et al.[9] found that polygenic risk scores (PRS) can independently influence the phenotype amongst carriers in several monogenic diseases. Finally, individual case reports have identified direct genetic epistatic modifiers, that is genetic background acting directly *through* the carrier variant's mechanism, that are protective in highly penetrant monogenic disorders[10]. While some studies[11] have identified digenic variants whose effects modify the impact of pathogenic variants, these studies did not identify whether genetic background variants directly interact with pathogenic variants, i.e., provide evidence of marginal epistasis, to affect penetrance and/or severity in biobank level data.

## Table 1 | Demographics and distributions of patients in the discovery cohort

| Participant information | |
| --- | --- |
| participants, *n* | 200,628 |
| female, *n* (%) | 110,475 (55.1%) |
| European ancestry, *n* (%) | 188,027 (93.7%) |
| Age at recruitment, mean (sd) | 56.5 (8.1) |
| HDL mg/dl, mean (sd) | 56.4 (14.8) |
| LDL mg/dl, mean (sd) | 145.9 (34.1) |
| triglycerides mg/dl, mean (sd) | 159.1 (94.7) |
| BMI kg/m², mean (sd) | 27.4 (4.8) |
| T2D, *n* (%) | 12,382 (6.2%) |

Analyses and data generated in this paper were performed on the 200k exome-sequencing release from UK Biobank cardiometabolic traits as our discovery cohort.

Here, we employ recently developed statistical genomics methods in combination with phenotypes and exomes from our discovery cohort of the 200,638 exomes release from UK Biobank (UKB)[12] participants (Table 1), as well as replication in the 28,817 participants from the Mt. Sinai Bio*Me* Biobank (Supplementary Data 1)[13] and the 454,787 UKB exomes release[14], to comprehensively study these factors in genes associated with monogenic cardiometabolic conditions: high LDL cholesterol (familial hypercholesterolemia), high HDL cholesterol

**Table 2 | Summary of clinical, monogenic conditions and curated variants**

| Condition (formal name) | Condition (shortened name) | Monogenic genes | Total curated, pathogenic variants | Total UKB 200k exomes carrier identified | Total noncarriers with exome and phenotype available |
|---|---|---|---|---|---|
| Familial hypobetalipoproteinemia | LDL-lowering | *PCSK9, APOB* | 63 | 341 | 190,832 |
| Familial hypercholesterolemia | High LDL | *LDLR, APOB* | 87 | 414 | 190,766 |
| Familial hyperalphalipoproteinemia | High HDL | *CETP* | 27 | 120 | 176,489 |
| Familial hypertriglyceridemia | High triglycerides | *APOA5, LPL* | 20 | 211 | 191,104 |
| Maturity-onset diabetes of the young | MODY | *HNF1A, HNF4A, GCK* | 73 | 128 | 191,623 |
| Monogenic obesity | Obesity | *MC4R* | 20 | 148 | 199,655 |

Heterozygous clinical variants that were previously validated across monogenic genes (referenced through the paper as "curated" variants) that affect cardiometabolic traits. The total number of curated pathogenic variant carriers identified in UKB exomes 200k release is summarized; some individuals identified carried the same curated, pathogenic variant. Additional information, such as variant effect and total number of carriers per variant is available in Supplementary Data 2.

(familial hyperalphalipoproteinemia), high triglycerides (familial hypertriglyceridemia), monogenic obesity, and maturity-onset diabetes of the young (MODY) (Table 2 and Supplementary Data 2). We also examine variants within this set of disease genes that are beneficial, such as variants that are LDL-lowering (familial hypobetalipoproteinemia) or protect against obesity. Using these biobanks, we identified individuals carrying at least one allele of these autosomal dominant pathogenic variants, whom we refer to as "carriers."

First, to study effect size heterogeneity of variants within monogenic genes, we leverage our recently developed method for variant pathogenicity prediction based on the ESM1b protein language model (Fig. 1B)[15,16]. The effect of rare missense variants in protein-coding genes are often classified as variants of uncertain significance (VUS), or grouped into coarse categories such as "pathogenic" or "benign"[5]. Of the 206,594 missense variants curated in ClinVar[17], 57.5% (118,864) are labeled as VUS as of November 2021[18]. Classification of VUSs is crucial for diagnoses and treatment of genetic disorders[19], but there is still a gap in methods to address this problem[20]. This critically limits studies of effect size heterogeneity as well as the prognostic power of genomic medicine for many patients[21]. Our model produces numerical scores for any possible amino acid change in any protein, which we demonstrate are tightly coupled to phenotype for many genes.

Next, to examine additive polygenic background effect (Fig. 1C), we employ polygenic risk scores (PRS), which combine variant effects from genome-wide association study (GWAS) loci, to measure the additional genetic load on the phenotypes (Y) included in this study[22]. We improve upon previous studies by binning individuals into finer-grained PRS quantiles to identify the threshold at which PRS-risk exceeds that of established clinical, pathogenic variants.

Finally, we employ our recent method, FAst Marginal Epistasis test (FAME)[23], that quantifies the impact of genetic epistasis, or genetic interactions, on modification of individual variant's effects (Fig. 1D). Previous methods have identified genome-wide genetic interactions[24] and genetic-by-environment interactions[25] that affect phenotype; however, we utilize this method to focus on identifying genetic interactions that directly modify the effect size of carrier status of pathogenic variants. With FAME, we previously showed that genetic background modifies the effect of many common GWAS variants, with epistatic effects sometimes exceeding marginal effects by an order of magnitude across diverse traits, and have replicated known marginal epistasis effects such as on gene expression[23,26]. Here, we extend this work to study the impact of marginal epistasis on autosomal dominant rare variants, i.e., identifying if genetic background variants are interacting with pathogenic variants to affect carrier phenotype and penetrance.

We find that the variant effect heterogeneity, additive polygenic risk, and marginal genetic epistasis each contribute to disease severity and penetrance in these traits. Importantly, a variant's ESM1b scores are predictive of phenotype severity in six out of ten monogenic genes (Table 2) included in this study. ESM1b outperforms other variant prediction methods for predicting clinical effect of monogenic missense variants even at rare allele frequencies and distinguishes between GOF and LOF missense variants. These results indicate that contemporary variant pathogenicity prediction methods extend beyond binary pathogenic/benign classification to provide more nuanced prognoses. We assessed the additive and epistatic effect of genetic background on the phenotype of carriers and found that PRS was significantly associated with phenotype severity for four of the six monogenic diseases examined in this study, demonstrating that polygenic background has an independent effect on carrier phenotype. In addition, we show that marginal epistasis, the effect of genetic background directly on the monogenic variant, significantly modifies the effect of the monogenic variant in carriers of high triglycerides, high LDL, and MODY variants. Inclusion of marginal epistasis in prediction of carrier phenotype could improve predictive accuracy by as much as 170%.

## Results

### Incomplete penetrance and variable disease severity of monogenic cardiometabolic variants

To establish the full spectrum of genetic contributions to "monogenic" diseases, we sought to determine the penetrance and disease severity across a subset of cardiometabolic traits within the UK Biobank (UKB). Cardiometabolic traits are pervasive quantitative phenotypes available within electronic health record (EHR) systems and have been previously associated with rare monogenic variants and common genetic variation. In the UKB, we identified a total of 1356 carriers of the curated monogenic variants that affect cardiometabolic phenotypes (Table 2 and Supplementary Data 2) and established that the penetrance for disease within these carriers is higher, but incomplete compared to disease prevalence within noncarriers using current clinical thresholds defined in the "Methods" (Fig. 2A). The monogenic trait with the highest penetrance was high triglycerides, where 56.10% (115/205) of carriers had triglycerides levels greater than 200 mg/dl; the monogenic trait with the lowest penetrance was for the LDL-lowering variants, where 42.28% (137/324) carriers had LDL levels less than 80 mg/dl.

Penetrance is also dependent on the gene that the variant was carried in; for example, penetrance of LDL-lowering variants (LDL < 80 mg/dl) was 42.28%, but was only 13.41% (22/164) in *PCSK9* pathogenic variants compared to 72.05% (116/161) in *APOB* pathogenic variants. Concomitantly, underlying phenotypes are variable amongst variant carriers of different genes (Fig. 2B). *GCK* MODY carriers have a narrower range of HbA1c, a measurement of blood glucose concentration[11], in comparison to *HNF1A* and *HNF4A* MODY carriers who have a wider range of values. Across traits and genes, this diversity of variant effect spans negligible to clinically actionable. We therefore examine the underlying factors that affect this incomplete penetrance and variable disease severity.

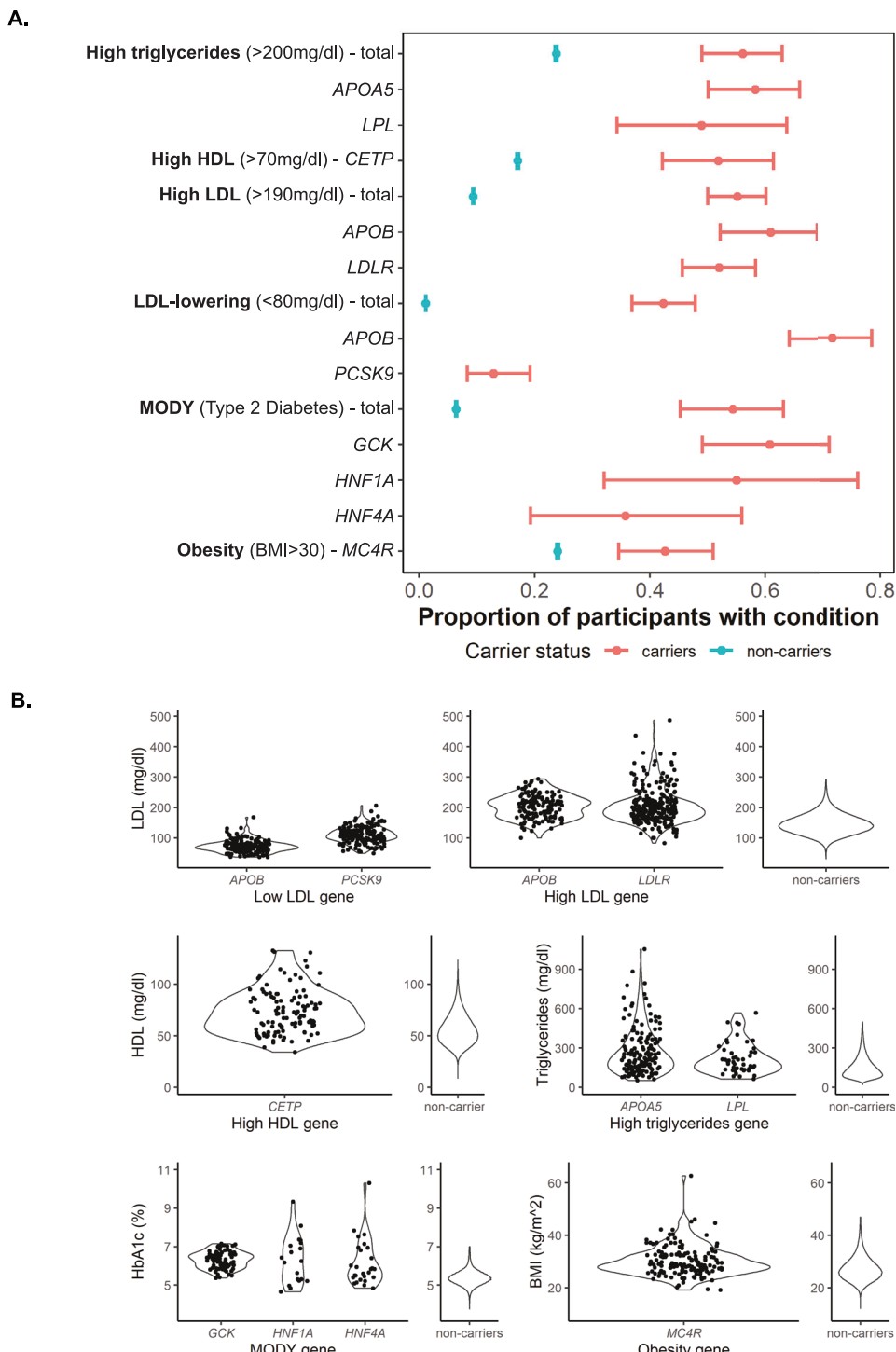

**Fig. 2 | Carriers of pathogenic variants that affect cardiometabolic traits have incomplete penetrance and variable disease severity. A** Penetrance thresholds were defined based on clinical definitions of disease. Relative to noncarriers (blue), carriers (pink) have higher rates of disease across all cardiometabolic phenotypes included in this study; error bars are based on 95% confidence intervals. Totals of individuals with phenotype recorded in table on the right. Carriers also show incomplete penetrance of disease across all monogenic disorders. **B** Among pathogenic variant carriers, we observe different severity of phenotypes. Total individuals (carriers and noncarriers) listed in Table 2; carrier totals based on gene of variant carried supplied in Supplementary Data 2.

## Severity of monogenic missense variants is predicted by ESM1b scores

We first consider the possibility that effect size heterogeneity across non-synonymous (missense) variants within a gene contributes to phenotypic heterogeneity of known autosomal dominant cardiometabolic traits; i.e., each pathogenic variant has each own respective effect size β (Table 2). There have been previous reports that different pathogenic variants within the same gene display differing disease penetrances[27–31] or expressivity[32]. We expand on this by employing ESM1b derived protein language scores[15] to predict the severity of missense variants across the 10 cardiometabolic genes. ESM1b defines likely pathogenic missense variants with a score less than −7.5[16]. While we and others have previously shown that variant pathogenicity predictors can help classify variants as pathogenic versus benign[16,33], we

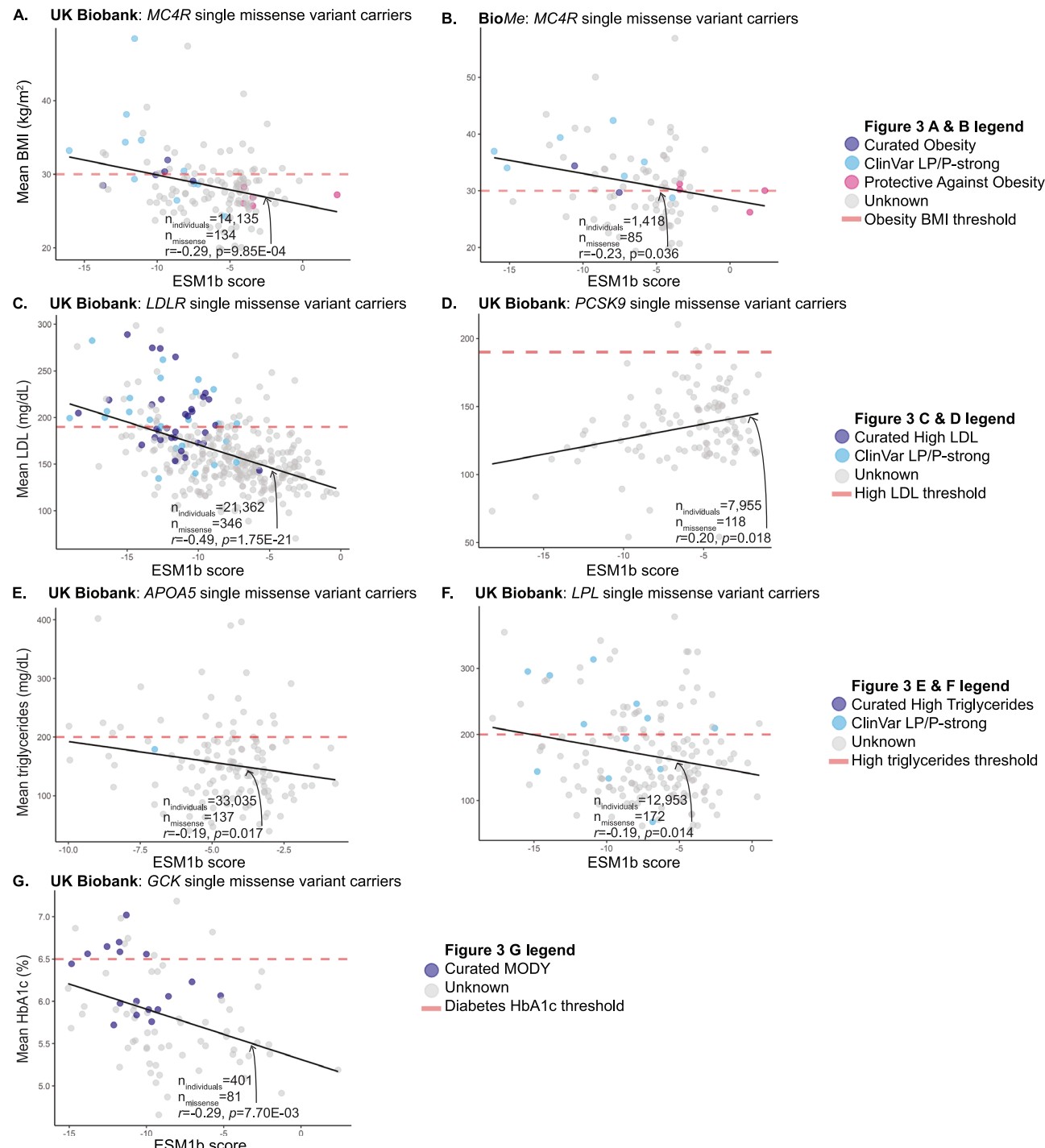

**Fig. 3 | ESM1b scores are predictive of disease severity for missense variant carriers.** Single missense variant carriers for *MC4R* (**A**, **B**), *LDLR* (**C**), *PCSK9* (**D**), *APOA5* (**E**), *LPL* (**F**), and *GCK* (**G**) were identified and mean phenotype per each missense carrier group was measured. *p* values shown are generated from mean phenotype-ESM1b score two-sided Pearson correlation tests and adjusted for age, sex, and first 10 genetic PCs. ESM1b scores also distinguish gain- vs. loss-of-function variants (**A**), which is replicated in Mt. Sinai's Bio*Me* biobank (**B**).

find that ESM1b predicts the mean phenotype of missense variant carriers with $p < 0.05$ for six of the ten genes considered (Fig. 3; binomial enrichment $p = 2.76E-06$). Two of these gene ESM1b-mean phenotype correlations are remarkably strong with correlations exceeding 0.25 and are significant after Bonferroni correction. Filtering to rarer variants further increases predictive power; an additional gene ESM1b-mean phenotype gains significance after filtering for rarer variants (Supplementary Data 3).

We next asked whether ESM1b could distinguish between LOF and GOF variants, something that none of the previous variant pathogenicity predictors have been able to do. We first explored *MC4R*, a single exon gene where missense variants have either LOF or GOF effects[7] leading to either monogenic obesity or protection from obesity, respectively. We identified carriers of both curated[4,34] and ClinVar-strong missense variants and quantified the association of these variants with their ESM1b scores. We found that ESM1b scores of these

known pathogenic missense variants are significantly associated with carrier BMI after adjusting for age, sex, and the first 10 genetic PCs in UKB (Pearson $r = -0.47$, $p = 0.034$). ESM1b also predicts phenotype in carriers of missense VUS (Fig. 3A), allowing for more accurate classification in the absence of molecular functional data. We extended our analysis to 14,135 individuals in UKB harboring any single missense variant in *MC4R* (134 unique missense variants). ESM1b score was significantly correlated with mean BMI of corresponding carriers after adjusting for covariates ($r = -0.29$, $p = 9.85E-04$). Finally, we found that ESM1b separates *MC4R* GOF (pink) from LOF (navy) missense variants (Fig. 3A); (*t*-test $p = 1.42E-04$). We replicated these results in an ancestrally diverse cohort of patients from the Bio*Me* biobank (Fig. 3B). In 1456 individuals that carry a single *MC4R* missense variant out of a total 28,817 individuals, ESM1b was significantly correlated with mean BMI ($r = -0.23$, $p = 0.036$).

We next examined ESM1b scores for *LDLR* and *PCSK9* missense variants in relationship to LDL levels (Fig. 3C, D). *LDLR* encodes for the LDL receptor; pathogenic/LOF variants account for 90% of monogenic high LDL cases[35] and disrupt LDLR's ability to remove LDL from the bloodstream leading to elevated LDL blood levels[36]. The ESM1b scores of known pathogenic missense variants are significantly associated with LDL after adjusting for age, sex, and first 10 genetic PCs ($n = 298$, $r = -0.46$, $p = 1.28E-3$). ESM1b accurately classifies the curated missense LOF variants (navy, Fig. 3C) as likely pathogenic; 23/24 (95.83%) had an ESM1b score $< -7.5$. Interestingly, the remaining pathogenic missense variant, with a score $> -7.5$, also had lower LDL levels compared to the other pathogenic missense variants. ESM1b was also able to predict phenotype in carriers of *LDLR* missense VUSs. In all 21,362 individuals carrying a single missense *LDLR* variant, representing 346 unique missense variants, ESM1b was significantly correlated with mean LDL ($r = -0.49$, $p = 1.75E-21$, Fig. 3C); these results also replicate in the Bio*Me* exomes ($r = -0.31$, $p = 3.65E-4$, $n_{missense} = 126$, $n_{individuals} = 3889$). We observed similar significant correlations between *PCSK9* missense variants and LDL levels, but in the opposite direction ($r = 0.20$, $p = 0.018$, Fig. 3D). Interestingly, there was no significant difference in LDL levels of carriers reported[37] *PCSK9* GOF and LOF variants (Fig. S2), highlighting complexities in reporting based on existing annotations[38,39].

Similar associations between ESM1b pathogenicity scores and phenotype were found in known clinical and VUS missense variants for additional genes and traits. *APOA5* and *LPL* LOF variants are associated with hypertriglyceridemia yet few missense variants are associated with these clinical phenotypes. We found that ESM1b scores are a predictor of triglyceride levels in missense variant carriers of both *APOA5* ($r = -0.19$, $p = 0.017$, Fig. 3E) and *LPL* ($r = -0.19$, $p = 0.014$, Fig. 3F). These correlations also replicate in the same direction and approach significance in Bio*Me* - *APOA5*: $r = -0.26$, $p = 0.066$. $n_{missense} = 50$, $n_{individuals} = 3049$; *LPL*: $r = -0.23$, $p = 0.11$, $n_{missense} = 52$, $n_{individuals} = 2016$. ESM1b scores also predicted HbA1c levels in *GCK* single missense variant carriers. *GCK* encodes for glucokinase, an enzyme that regulates insulin secretion[40]. Variation in *GCK* has been associated with both hyperglycemia and hypoglycemia[41]. ESM1b predicted the mean HbA1c levels of 401 single *GCK* missense variant carriers in Fig. 3G ($r = -0.29$, $p = 7.7E-03$).

To assess whether other variant effect predictors had the same features as ESM1b, we repeated these analyses using SIFT[42], CADD[43], PolyPhen2[44], PrimateAI[45], AlphaMissense[46], and EVE[47] scores and found that these methods do not classify the pathogenic missense variants as accurately as ESM1b, show weaker correlations between variant score and mean BMI compared to ESM1b, and do not differentiate between GOF and LOF missense variants (Fig. S1 and Supplementary Data 3).

We also found that ESM1b scores remain predictive of carrier phenotype at missense variants with small allele frequencies (Supplementary Data 3). We replicate these results for five of the six phenotype correlations in additional individuals within the 500k UKB

exomes, excluding individuals already present in the 200k exomes (Fig. S3); the remaining phenotype correlation approaches significance ($p = 0.0666$). Collectively, these results suggest that effect sizes of clinical variants within a gene are heterogeneous and therefore contribute to variability in penetrance and disease severity. They also indicate that ESM1b has the potential to reclassify thousands of variants that have conflicting classifications or are of uncertain significance.

## Polygenic background in carriers and non-carriers of pathogenic variants

Next, we addressed another source of phenotypic heterogeneity amongst carriers of the same pathogenic genetic variant using tools such as polygenic risk scores (PRS), a weighted sum of common variant effects with weights determined by results from GWASs[48], and emerging large scale biobanks (Fig. 1C) for each trait of interest (Table 2). Previous studies have shown that polygenic background additively affects disease severity[4,8] in rare variant carriers across a variety of traits. We leverage a larger, more powered release of UKB to investigate PRS and pathogenic variants, restricting to the unrelated white British population to reduce confounding from population structure[49] (see "Methods").

Consistent with previous studies, each PRS was significantly correlated with the corresponding traits in carriers (Fig. S4). Then, to compare polygenic and monogenic risk, we contrast the phenotypes of noncarriers within the tails of 1000th-tiles (0.1%) bins of the PRS to the phenotypes of pathogenic variant carriers to identify the exact percentile where noncarriers have more extreme phenotypes than carriers. We tested PRS for non-carriers for monogenic obesity, HDL and triglycerides and find that individuals in the tails of PRS have more extreme phenotypes than individuals in the tails of PRS for HDL and triglycerides have phenotypes larger than individuals harboring curated clinical variant carriers (Fig. 4A-C). Across all three traits we observe that hundreds to thousands of individuals have a polygenic load that results in a more extreme phenotype than currently reported clinical variants. Exact PRS percentiles at which non-carrier phenotypes exceed those of carriers are reported in Supplementary Data 4 and are denoted in red in Figs. 4 and S5. These findings replicate that individuals within the tails of PRSs are at equivalent or greater risk of disease than pathogenic variant carriers[4,50]. While individuals in the tails of the current LDL and Type 2 Diabetes (T2D) PRS do not have phenotypes exceeding those of clinical variant carriers, this will likely change as PRS become more accurate and larger cohorts are studied. We also replicated Ripatti et al.'s[51] work in additional phenotypes and observed an enrichment of noncarriers with extreme PRSs within individuals that meet disease thresholds (Supplementary Data 5).

We examined several different sets of potentially pathogenic variants when making these comparisons: a curated set of variants (Table 2 and Supplementary Data 2), ClinVar-weak/strong annotations (see "Methods"), and VUSs with ESM1b scores exceeding the recommended pathogenicity threshold of −7.5 (see "Methods"). For all traits examined, the curated variants had the most extreme phenotypes while carriers of ClinVar's current set of weak and strong variants often had substantially more moderate phenotypes (Fig. 4A, C, D). ClinVar variants for LDL did not distinguish between increasing or lowering LDL effects and therefore were not included in Fig. 4D. We found that ESM1b could be used to identify additional pathogenic variants: ESM1b annotated pathogenic VUS missense variants had phenotypes equivalent to or more severe than ClinVar variant carriers for some genes (Fig. 4A, C).

Finally, we examined the independent effect of polygenic background in carriers of clinical variants for cardiometabolic disease. Studies of other traits have reported correlations between PRS and phenotypes amongst rare monogenic disease variant carriers[8,52–54]. In monogenic forms of cardiometabolic disease, this association has not

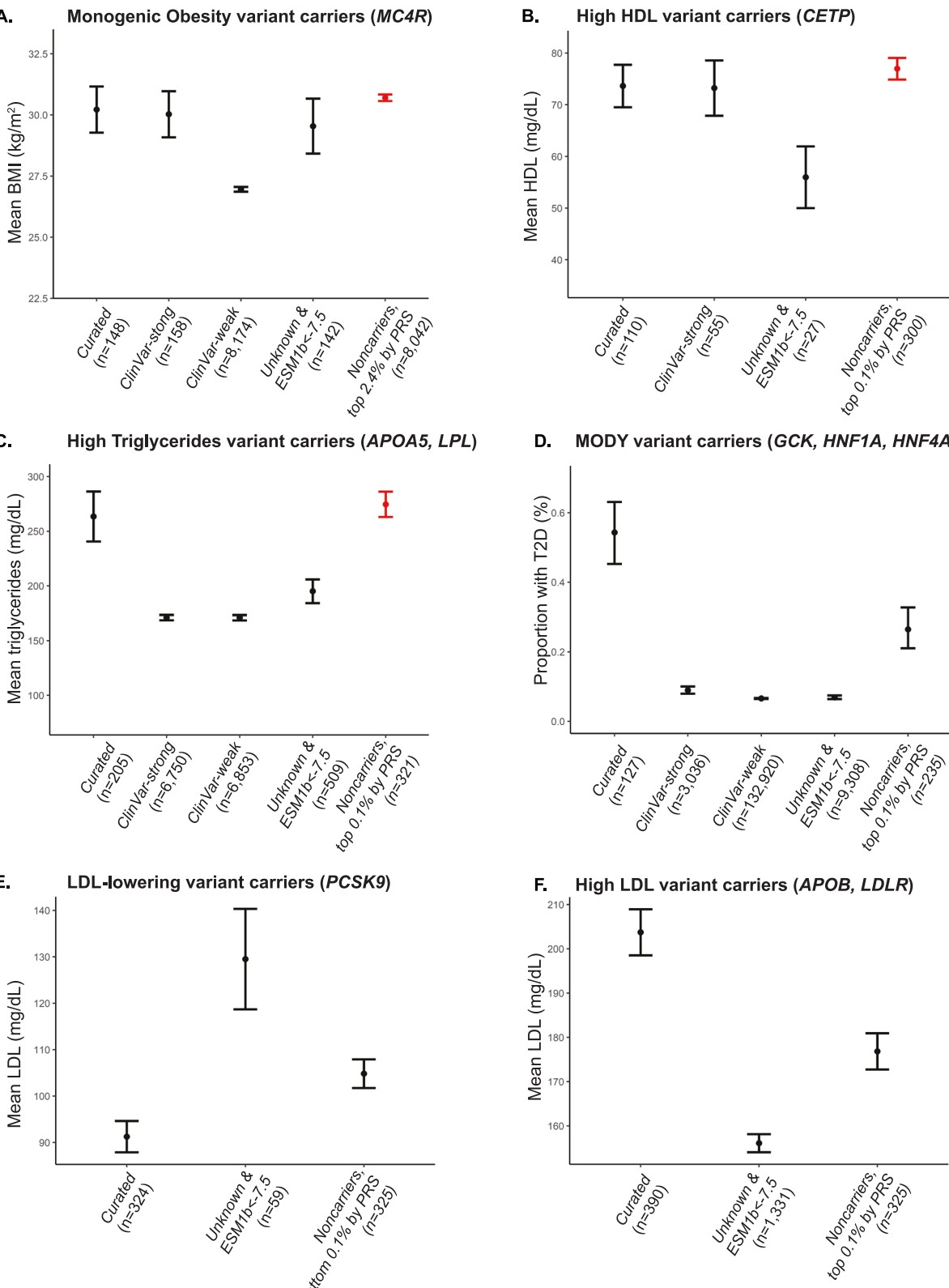

**Fig. 4 | Comparison of phenotypic distributions amongst differential defini-tions of carrier status and tails of noncarrier PRSs.** Red indicates non-carriers with mean phenotype equivalent or more extreme than curated pathogenic variant carriers: monogenic obesity (**A**), high HDL (**B**), and high triglycerides (**C**). Only MODY (**D**), LDL-lowering (**E**) and high LDL (**F**) curated pathogenic variant carriers had more extreme phenotypes than noncarriers in PRS tails. Error bars are based on 95% confidence intervals. Additional carriers were identified by using ClinVar and

ESM1b. Pathogenic/likely pathogenic ClinVar variants in monogenic genes were identified with different filtering stringency ("weak"−less stringent filtering, "strong"−more stringent filtering), and potentially pathogenic missense variants with unknown function were identified with ESM1b < −7.5. ClinVar variants are not included in (**E**) and (**F**) because pathogenic variants were not denoted as high LDL or low LDL effect.

been established due to insufficient sample size[4]. Here, we found that carrier phenotype was significantly associated (Bonferroni-corrected, one-tail $p$ value < 0.01) with carrier PRS while adjusting for carrier sex, age, and first 10 genetic PCs in monogenic obesity ($\beta = 1.68$, $p = 5.60\mathrm{E}{-03}$), high HDL ($\beta = 9.79$, $p = 1.57\mathrm{E}{-06}$), LDL-lowering ($\beta = 9.87$, $p = 3.18\mathrm{E}{-06}$), and high triglycerides ($\beta = 62.46$, $p = 1.33\mathrm{E}{-05}$) carriers (Fig. S4A–C, E). LDL PRS approached significance in high LDL carriers ($\beta = 6.76$, $p = 0.028$, Fig. S4D). For MODY carriers, we predicted T2D status using a logistic regression including T2D PRS, age, sex, and the first 10 genetic PCs as covariates; the T2D PRS covariate was not significant ($\beta = 0.44$, $p = 0.15$). The PRS covariate for all sets of monogenic carriers is positive, indicating that the higher the carrier PRS is, the larger the value of the carrier phenotype. Additionally, we adjusted for PRS in unrelated, European individuals carrying missense variants in monogenic genes to determine if this improved correlation results (Supplementary Data 3); we found improvement in significance of the correlation. Across all traits, our results support previous findings that polygenic background is a source of incomplete penetrance and variable disease severity and add well powered studies of cardiometabolic phenotypes that demonstrate the effect of the additive effect of PRS to phenotype expression in additional monogenic disorders.

## Marginal epistasis between genetic background and monogenic genes alters phenotype

We next sought to evaluate the possibility that genetic background magnifies or diminishes the effect size of the pathogenic variants through marginal epistasis (Fig. 1D)[10,55–57]. This notion of interaction is termed marginal epistasis[58]. One of the major challenges in identifying marginal epistasis is the computational bottleneck of testing all genetic interactions at scale within hundreds of thousands of samples in a biobank. To do this, we employed a novel mixed model based approach (FAME)[23] that estimates the total contribution to phenotypic variance from polygenic background ($\sigma_G^2$), carrier status ($\beta_C^2$), their interaction ($\sigma_{C \times G}^2$), and environmental noise ($\sigma_e^2$). This allowed us to conduct the first well-powered examination of the impact of marginal epistasis on penetrance and disease severity. While others have tested for gene-environment interactions[25] and all pairwise genome-wide interactions that influence phenotypes[24] (whose estimators have large standard errors and low power), we solely focus on identifying the common genetic variation that is interacting with carrier status to modify phenotype. We note that testing for PRS-carrier status interaction is an underpowered version of our approach with very limiting assumptions; we applied this underpowered test and did not identify novel interactions (Supplementary Data 6, Supplementary Methods).

In the FAME model, $\sigma_G^2$ is the phenotypic variance explained by genetic background and represents the theoretical upper limit of polygenic risk score accuracy for each trait. $\beta_C^2$ is the variance explained by carrier status and $\sigma_{C \times G}^2$ is that variance explained by marginal epistasis between carrier status and genetic background. Here, we compute the epistatic improvement percentage, $EIP = 100 * \sigma_{C \times G}^2 / \beta_C^2$, which is the ratio between marginal epistasis variance and carrier status variance. It represents the upper bound of

improvement in phenotype prediction over carrier status that can be achieved through modeling marginal epistasis. An $EIP$ of 0% means that epistasis is not present, while an $EIP$ of 100% means that the combined epistatic effects on the pathogenic variant are as large as the direct pathogenic variant effect and marginal epistasis is a substantial factor modifying phenotype amongst carriers.

Our analyses revealed widespread statistical evidence of marginal epistasis with large effect sizes; $EIP$ ranged from 48% to 170% amongst the significant associations after Bonferroni corrections (Table 3 and Supplementary Data 7). $EIP$ was 170% (standard error: 33.35%) for LDL cholesterol (interaction $p = 1.2\mathrm{E}{-08}$), implying that an ideal model including epistasis would be 2.7 times more accurate in predicting cholesterol compared to using carrier status alone. The fact that $EIP$s exceed 100% suggest that marginal epistasis is a substantial contributor to variable penetrance and disease severity. These modifications could act through a variety of mechanisms including eQTLs modifying the expression levels of the monogenic gene[59], disruptions to enhancer sequences that affect the monogenic gene transcription[60], and alternative splicing of proteins that interact with monogenic genes[57]. Identifying the loci and pathways involved in these marginal epistatic interactions could also reveal opportunities for treatment. We caution that, like all tests of gene-gene and gene-environment interaction, endogeneity and scale can induce biases in effect size estimates.

## Discussion

The question of why some monogenic variant carriers have extreme phenotypes while others remain healthy is fundamental to clinical genetics. In this study, we established at biobank scale three genetic contributors to phenotypic heterogeneity of pathogenic variant carriers: differing effect sizes of missense variants in a monogenic gene, genetic background independently affecting carrier phenotype, and marginal genetic epistasis modifying phenotype through direct effect on the variant. Our study provides clarity on how rare and common genetic variants can have independent effects and interact to modify the phenotype severity. Importantly, this work lays a foundation for improved prognostic ability by incorporating complete genomic information in clinical interpretation.

There remain a few limitations to our study. Most clinical pipelines define the canonical isoform as the longest protein-coding transcript[61] or use MANE-defined transcripts[62]. However, the cell-type specific isoforms[63], the importance of multiple clinically relevant isoforms[64] and the ratios of these isoforms[65] are understudied areas of variation that can be probed using long-read sequencing technologies. Furthermore, each gene and disease phenotype have different contributions from rare variants and genetic background to an individual's phenotype requiring large and well-curated data sets across diverse populations to establish the contributors to phenotype disease severity and penetrance.

The measured penetrance of pathogenic variants drifts over time with revisions of screening guidelines, diagnostic thresholds and improved therapies. Our measures of penetrance may also be affected

## Table 3 | Marginal epistasis with monogenic genes results

| Trait | Monogenic genes tested | $\sigma_{C \times G}^2$ | $\beta_C^2$ | $EIP$ (%) | $EIP$ SE | $p$ |
|---|---|---|---|---|---|---|
| LDL | High LDL (*APOB, LDLR*)* | 2.92E−03 | 6.07E−03 | 48.02 | 10.66 | 2.88E−10 |
| Triglycerides | High triglycerides (*APOA5, LPL*)* | 2.89E−03 | 1.68E−03 | 172.36 | 33.35 | 1.22E−08 |
| HDL | High HDL (*CETP*) | 5.40E−04 | 8.74E−04 | 61.75 | 30.65 | 0.010 |
| HbA1c | MODY (*GCK, HNF1A, HNF4A*)* | 9.21E−04 | 1.58E−03 | 58.17 | 21.82 | 3.60E−04 |

After adjusting for age, sex, and the first 20 genetic PCs, the interaction term between background variation and carrier status remained significant for High Triglycerides carriers, High LDL carriers, and MODY carriers (significant monogenic genes after Bonferroni corrections marked with *, less than $p = 0.05/6 = 0.0083$). We show the proportion of variance in phenotype across carriers and noncarriers modulated by marginal epistasis ($\sigma_{C \times G}^2$), due to carrier status ($\beta_C^2$), and the ratio between $\sigma_{C \times G}^2$ and $\beta_C^2$ (epistatic improvement percentage, $EIP$). $EIP$ represents the potential improvement in carrier phenotype prediction when modeling epistasis. Marginal epistatic interactions between common background variation and carrier status were tested using the FAME method.

by our variant calling process. Because we did not verify variant calls via inspection of CRAM files, some individuals identified may not actually carry the variants of interest and are false positives that affect our results. Like polygenic risk scores, results can vary based on thresholds used to distinguish between healthy and disease states. For cardiometabolic disorders, there are many medications that improve lipid profiles, such as statins[66], and our study adjusted for statin-usage and predicted pre-medication LDL and triglyceride levels utilizing coefficients that were previously calculated[67,68]. However, there are many different statins and likely each of these have not only dosage- but also genetically-driven responses to drug therapy[69]. Finally, newer drugs for obesity and the rise of procedures such as gastric bypass surgery, are artificially reducing BMI and improving lipid profiles[70,71] and, over time, may significantly decrease estimates of penetrance and disease severity of metabolic traits.

Within this study, we take advantage of quantitative traits associated with pathogenic variants to study factors that affect disease severity within carriers. This disease severity is a limited proxy for expressivity. Clinical expressivity is often used with an alternate definition referring to different phenotypes that arise from individuals carrying the same pathogenic variant. Studying this type of expressivity is essential, but will require a priori knowledge of the full spectrum of the clinical phenotypes possible, a structured database for these phenotypes within a biobank. Even the largest biobanks may be underpowered, particularly when relying on EHRs, where absence of the phenotype in records is not an indication of the patient being unaffected.

Going forward, examination of our findings across global populations is essential, but will require diverse large-scale biobanks with exome sequences linked with clinical phenotypes. While the effect of the isolated pathogenic carrier variants is currently believed to be consistent, we and others have observed that heterogeneity of clinical expression is influenced by genetic background, which differs between populations. VUS are more common in non-European populations for many disease genes[72] and exome sequencing analysis that takes into account diverse genetic backgrounds will remedy this problem[72,73]. Finally, extension into other phenotypes will be most successful for quantitative traits that are measured in the majority of a biobank's participants. These hurdles will differ between phenotypes assessed and across biobanks.

In addition to providing a means of studying variable penetrance and disease severity, the ESM1b analyses resulted in discoveries with translational potential for the interpretation of clinically observed genomic variants. Integration of precision genome medicine into routine clinical care requires improved variant pathogenicity prediction models. Early methods[42,43] show diminished variant pathogenicity prediction accuracy as they rely on an imperfect and underpowered "gold-standard" truth set. Newer methods, such as ESM1b, AlphaMissense[46], and PrimateAI-3D[33], are based on improved machine learning methods and have increased pathogenicity prediction accuracy. ESM1b[15,16] is a 650 million parameter protein language model trained on 250 million protein sequences that can predict which variants are pathogenic at higher accuracy than existing variant pathogenicity prediction models, provide scores that correlate with a continuous spectrum of clinical phenotypes, and is freely accessible online[15,16]. Evaluating variant pathogenicity methods via large-scale biobanks allows us to assess the accuracy of these predictors in clinical environments, expanding beyond in vitro functional analysis, and previously published cases that are biased towards the most severe phenotypes. Our results show that ESM1b outperforms other variant pathogenicity predictors in two clinically significant ways: first, it can classify established pathogenic variants and variants across a continuous range of effect sizes, and second, it distinguishes between GOF and LOF missense variants. A previous analysis of rare variation pathogenicity using PrimateAI-3D[33] shares some common findings

with this study. However, it focused on incorporation of scores to quantify rare variant polygenic risk rather than understanding penetrance and disease severity[74].

In summary, our study provided evidence of reduced penetrance in a large population cohort and discovered how genetic background can have outsized effects on modulating rare-variant clinical prediction. It also established a contribution of both rare, monogenic effects and the influence of a polygenic background on the clinical phenotype. Our work highlights the critical importance of the integration of rare and common variants and how these have the power to improve clinical prognosis of genomic precision medicine.

## Methods
### Cohort information
All research completed with approved UK Biobank and BioMe Biobank applications. A total of 200,632 participants with exomes in UKB[12] were included to identify the number of carriers and the penetrance of the monogenic diseases in this study. We restricted PRS and genetic, marginal epistasis analyses to individuals of similar genetic ancestry who are unrelated. Field 22006 was used to identify individuals who both self-identify as White British and have similar genetic ancestry based on PCA. To identify unrelated individuals, common array SNPs were extracted from individuals, KING[26] kinship coefficients were estimated, and individuals were pruned to the third degree of kinship. All individuals with exomes available were included in the missense variant analysis.

The BioMe biobank is an electronic health record-linked biorepository that has been enrolling participants from across the Mount Sinai health system in NYC since 2007. There are currently over 50,000 participants enrolled in the BioMe biobank under an Institutional Review Board (IRB)-approved study protocol and consent. Recruitment occurs predominantly through ambulatory care practices, and participants consent to provide whole blood-derived germline DNA and plasma samples which are banked for future research. Participants also complete a questionnaire providing personal and family history as well as demographic and lifestyle information as has been previously described[75,76]. BioMe participants represent the broad diversity of the New York metropolitan area, and more than 65% of participants represent minority populations in the US. All participants provided informed consent, and the study was approved by the Icahn School of Medicine at Mount Sinai's IRB (#07-0529).

### Cardiometabolic phenotype ascertainment
Direct LDL, HDL, and triglycerides (respectively, fields 30780, 30760, 30870; mmol/L) for each participant were obtained and converted to mg/dL from mmol/L (LDL & HDL: multiplied by 38.67; triglycerides multiplied by 88.57)[77]. The mean of these measurements taken across multiple visits were used to represent each individual. LDL and triglycerides measures were adjusted to account for patient statin use; for LDL, patient's measurement was divided by 0.7 and triglyceride was divided by 0.85[67,78].

Maximum body mass index (BMI, $kg/m^2$, field 21001) recorded was used to represent each individual. A BMI between $25\,kg/m^2$ and $30\,kg/m^2$ was considered overweight; a BMI greater than or equal to $30\,kg/m^2$ defined obese participants.

Glycated hemoglobin or HbA1c (field 30750; mml/mol) and the "Diabetes diagnosed by doctor" field were used to identify participants with Type 2 diabetes (T2D). HbA1c was converted from mmol/mol to percentage, and the maximum HbA1c measured across all instances was used to represent each individual. Participants were identified as having T2D if they fulfilled at least one of the following criteria: (1) HbA1c greater than or equal to 6.5%, (2) at least one instance of "Diabetes diagnosed by doctor" marked TRUE. Participants were identified as being pre-diabetic if their HbA1c was between 5.7% and 6.5%.

## Gene and variant list curation

There are several terms used interchangeably to describe variants that have high effect and are associated with monogenic disease (e.g., "pathogenic", "monogenic", "clinical"). We focus on pathogenic variants as defined by ACMG/AMP criteria[68]. We examined pathogenic variants for monogenic forms of low LDL-lowering or familial hypobetalipoproteinemia (*PCSK9, APOB*), high LDL or familial hypercholesterolemia (*LDLR, APOB*), high HDL or familial hyperalphalipoproteinemia (*CETP*), high triglycerides or familial hypertriglyceridemia (*APOA5, LPL*), monogenic obesity (*MC4R*), MODY (*GCK, HNF1A, HNF4A*) curated in Goodrich et al.[4] and Mirshahi et al.[61] (Table 2 and Supplementary Data 2). Any person carrying at least one allele of these pathogenic variants will be referred to throughout this text as a "carrier". We consider several classes of variants to identify monogenic variant carriers: "curated", where variants undergo stringent review to be considered pathogenic; "ClinVar-weak", where variants have at least one submission of likely pathogenic or pathogenic, but may also contain conflicting reviews in the ClinVar database[17]; and "ClinVar-strong", where variants have only likely pathogenic or pathogenic submissions. Variants that did not fall under "ClinVar-strong" or "curated" categories were considered to be variants of uncertain significance (VUS). Supplementary Methods Table 1 summarizes these definitions.

"Curated" monogenic variants were identified by applying ACMG/AMP criteria and blinded testing by reviewers for variant curation by Goodrich et al.[4] and Mirshahi et al.[61]. Rare protein-truncating variants in *HNF1A*, *HNF4A*, and *GCK* outside of the last exon of each gene were classified as pathogenic due to haploinsufficiency of these genes is sufficient to cause disease. Missense variants within these genes were also identified as pathogenic for MODY if the missense variants were classified as likely pathogenic/pathogenic by ACMG/AMP guideline, were rare (minor allele frequency, MAF < 1.4E−05), and were also subjected to blinded manual review. ClinVar variants were identified based on the "CLIN_SIG" field from the Variant Effect Predictor (VEP)[34].

## Exome sequencing quality control and variant filtering

UKB exome-sequencing and analysis protocols were published in Szustakowski et al.[79] and are also displayed at https://biobank.ctsu.ox.ac.uk/showcase/label.cgi?id=170. Exome variants were called in monogenic disease genes by using PLINK version 1.9 function *extract* on UKB exome PLINK files[80]. Anyone carrying at least one pathogenic variant was identified as a "carrier"; otherwise, those not carrying pathogenic variants were labeled as "non-carriers". All variants were annotated using Variant Effect Predictor (VEP) version 107[34] in GRCh38.

## Penetrance calculations

We define penetrance as the proportion of carriers that meet certain disease or phenotype thresholds based on previous studies. In MODY carriers, penetrance was based on how many carriers had diabetes. For the other monogenic disorders, the following cutoffs were used to calculate penetrance: high LDL or familial hypercholesterolemia−direct LDL greater or equal to 190 mg/dl[81], LDL-lowering or familial hypobetalipoproteinemia−direct LDL less than or equal to 80 mg/dl[82], high HDL or familial hyperalphalipoproteinemia−direct HDL greater than or equal to 70 mg/dl[36], high triglycerides or familial hypertriglyceridemia−direct triglycerides greater than or equal to 200 mg/dl[81], and monogenic obesity−obese BMI (BMI greater or equal to 30 kg/m$^2$.)

## Missense variant pathogenicity prediction scores

ESM1b is a 650 million parameter protein language model that was previously trained on all 250 million protein amino acid sequences[15] in UniProt[83]. This unsupervised model is not trained on any genetic information or any other protein information outside of amino acid sequence. The model can predict the likelihood of any potential single amino acid change (missense variants) by calculating a score for the missense variant as the log likelihood ratio in comparison to the wildtype variant[16]. The ESM1b model was used to calculate the scores for any single amino acid change for the protein resulting from the canonical transcript of the monogenic disease genes included in this study. Here, we define the canonical transcript as the MANE-defined transcript[84]. Using the predicted protein change of the genetic variant effect generated by VEP, we compared the ESM1b scores for every potential missense variant of established cardiometabolic disease genes to the phenotypes of carriers for those missense variants.

We tested if ESM1b predicts mean phenotype of carriers of the same missense variants for all genes included in this study, restricting this analysis to single missense variant carriers from any ancestry. We define single missense variant carriers as individuals with one missense variant in the gene, and any other gene variation is restricted to intronic, synonymous, or untranslated region effects. Single missense variant carriers were grouped by the missense variant carried, mean phenotype of this group was measured and associated with the missense variant's ESM1b score. We then identified significant Pearson correlations between mean phenotype and ESM1b score via correlation testing; to account for covariates, we regressed age, sex, and the first 10 genetic PCs from the phenotype and then used the remaining residuals to test for correlation with ESM1b values. These correlations were replicated in the UKB 500k exomes release[14] by analyzing individuals within the new release only and excluding individuals in the 200k release (Fig. S3).

## Polygenic risk scores (PRS)

PRS weights for BMI were previously generated using LDpred[62] and were downloaded from Cardiovascular Disease KP Datasets on Feb 10, 2022. PRS weights for LDL were previously generated using PRS-CS[85] and were downloaded Feb 22, 2022 from the Global Lipids Genetics Consortium Results. PRS weights for HDL and triglycerides were previously generated using PRS-CS[86] and downloaded from the PRS Catalog[87] on May 6, 2022. PRS weights for T2D were previously generated using LDpred[88] and were downloaded from the PRS Catalog on May 29, 2023. PRSs were then calculated for every UKB participant of European ancestry within UKB using PLINK version 2.0 function *score*. Scores were then centered and scaled to have a mean of 0 and standard deviation of 1. All PRS weights chosen excluded UKB participants in generation of GWAS training data.

## Marginal epistasis to identify interaction between genetic background with monogenic gene variants

Testing for genetic epistasis, or gene-by-gene interactions, is a challenging task that is computationally expensive to scale in large datasets like biobanks. FAst Marginal Epistasis Estimation (FAME) is a scalable method that tests for marginal epistasis: how an individual's genetic background measured across hundreds of thousands of common genetic variants interacts with their carrier status to ultimately influence the trait[23]. Rather than a linear-regression model which measures the independent and additive effect of genetic background, in the form of PRS, FAME jointly estimates the variance explained by the additive component ($\sigma_G^2$) and by the marginal epistasis component ($\sigma_{C \times G}^2$), where the marginal epistasis is defined as the pairwise interaction between the target feature, and all other SNPs of interest. The algorithm for fitting these variance components in FAME is based on a streaming randomized method-of-moments estimator that has a runtime that has a linear scaling with the number of SNPs and individuals[89,90]. FAME also efficiently estimates asymptotic standard errors for the variance component estimates. While the original implementation of FAME was designed for testing marginal epistasis at common variants, we modified the FAME software to take as input the carrier status at the target gene (t) of interest ($C_t$), and genotypes that

potentially interact with the target feature ($G_t$). We partition the set of common SNPs into those that are proximal to the target gene of interest and those that are distal leading to corresponding genotype matrices, $G_1$ and $G_2$ respectively. We aim to test for interactions between carrier status and SNPs that are distal to the target gene (while controlling for additive effects across all common SNPs, additive effects of the carrier status, and relevant covariates).

When we estimated marginal epistasis for the pathogenic variants at a target gene, we first excluded the additive effect of carrier status together with the other covariates (top 20 PCs, sex, and age). Then we applied FAME to jointly estimate the additive SNP effect and the marginal epistasis effect on 119,523 unrelated White-British individuals with genotyping arrays and exome-sequencing available in the UKB. We have included more detailed FAME information in the Supplementary Methods and verified that linkage disequilibrium (LD) has little to no effect on our results in Fig. S6.

### Reporting summary
Further information on research design is available in the Nature Portfolio Reporting Summary linked to this article.

## Data availability
Both the UK Biobank and the Bio*Me* Biobanks are restricted due to patient privacy and are only available under an application process. UK Biobank access was obtained via https://www.ukbiobank.ac.uk/enable-your-research. BioMe access was obtained via requests submitted to Bio*Me* Biobank and Mount Sinai Data Warehouse. Databases also used in this work include: ClinVar (https://www.ncbi.nlm.nih.gov/clinvar/), gnomAD exomes v2.1 (https://gnomad.broadinstitute.org/), Cardiovascular Disease KP genetic association datasets (https://cvd.hugeamp.org/datasets.html), Global Lipids Genetics Consortium Results (https://csg.sph.umich.edu/willer/public/glgc-lipids2021/), PubMed (https://pubmed.ncbi.nlm.nih.gov/), and GoogleScholar (https://scholar.google.com/).

## Code availability
Software used is cited in the "Methods" section and all are open source: Plink v1.9 & 2.0, R v4.1.1, Ensembl Variant Effect Predictor v107, FAME v1.0. Code available in GitHub at https://github.com/angela-wei/penetrance_expressivity[91].

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

## Acknowledgements

This research has been conducted using UK Biobank data under application 33127 and is available through the UK Biobank Access Management System http://amsportal.ukbiobank.ac.uk/. Figure 1 created in BioRender: Arboleda (2025) https://BioRender.com/hmwost8. This work was supported in part through the computational and data resources and staff expertise provided by Scientific Computing and Data at the Icahn School of Medicine at Mount Sinai and supported by the Clinical and Translational Science Awards (CTSA) grant UL1TR004419 from the National Center for Advancing Translational Sciences. Research reported in this publication was also supported by the Office of Research Infrastructure of the National Institutes of Health under award number S10OD026880 and S10OD030463. The content is solely the responsibility of the authors and does not necessarily represent the official views of the National Institutes of Health. This work was supported by the following funding sources awarded to V.A.A., N.Z., and E.E.K.: R01HG011345. This work was supported by the following funding sources awarded to A.W.: F31HG013462.

## Author contributions

A.W., N.Z., and V.A.A. conceptualized the project and designed all experimental approaches. A.W., N.Z. and V.A.A. wrote and edited the manuscript with input from all authors. A.W. performed all computational experiments, curated all data—in addition to supervising and managing all components of this study. R.B. curated the UKB phenotypes and completed QC analyses. N.B. and V.N. ran the ESM1b model and provided ESM1b scores for missense carrier phenotype analysis. S.K.J. provided variant annotations. S.S. and B.F. designed and executed all computational analyses related to FAST epistasis analysis. E.E.K. provided access to BioMe exomes and S.C. identified single MC4R missense carriers. M.S.U. advised best practices for analyses and contributed to manuscript editing.

## Competing interests

E.E.K. has received personal fees from Regeneron Pharmaceuticals, 23andMe, Allelica and Illumina; has received research funding from Allelica; and serves on the advisory boards for Encompass Biosciences, Overtone and Galatea Bio. All other authors declare no competing interests.
