## [Transparent Peer Review file · Nature Communications]

Investigating the sources of variable impact of pathogenic variants in monogenic metabolic conditions

Corresponding Author: Dr VALERIE ARBOLEDA

Version 0:

Reviewer comments:

Reviewer #1

(Remarks to the Author)

This paper leverages the powerful datasets of the UKBb and the BioMe to examine the genetic basis of variable penetrance of monogenic effects on five cardiometabolic quantitative traits, triglycerides, LDL, HDL, BMI and HbA1c (MODY variants). First, the effect magnitude of the specific monogenic variant is quantified by ESM1b, an interesting novel method of pathogenicity prediction based on protein language that the authors have recently validated (Ref 14). ESM1B scores explain from 3.6% (APOA5 and LPL for triglycerides) to 24% (LDLR for LDL) of the quantitative trait variance (based on the r^2 values from Fig. 4), leaving a lot of room for polygenic effects in trans. These are first addressed by examining the effect of polygenic risk scores on the phenotype by showing, unsurprisingly, that complex-trait GRS values in non-carriers explain as much or more of the extreme phenotypic variance in these quantitative traits. The question of their relative contributions is addressed by FAME, a novel computational approach that calculates the EIP, which is shown, for four of the traits, to add very substantially to the prediction based solely on the monogenic variant.

This study confirms and systematizes, with repository-sized samples, previous work on trans effects on the penetrance or (more correctly for quantitative traits) expressivity of monogenic variants and is of potential clinical utility in determining pathogenicity of monogenic variants. It is a substantial contribution to the field. However, I find that, as explained below, the paper implicitly makes an important but unsupported claim. Also the paper will be much easier to read and understand, if the term “epistasis” with or without the qualifiers “marginal” and “genetic” is used in a rigorously defined and consistent matter.

Major concerns

1. Semantics: The term “epistasis” has been used by different authors to mean different things (reviewed by Phillips Nat Rev Genet. 2008 Nov; 9(11): 855–867, PMID: 18852697, doi 10.1038/nrg2452). By itself, it only means “genetic modifier(s) in trans”, and any other use should be rigorously defined or (better in my opinion) completely avoided and replaced by terms more meaningful in 21st-century genetics. The paper attempts (line 270) to define “marginal epistasis” as the $\sigma_{C \times G}$ component, i.e. clearly the effect due to non-additive interaction of complex-trait loci with the “target feature” (a jargon term for “monogenic variant”, I presume) at the express exclusion of σ_G , i.e. the simply additive effect of known GWAS loci as quantified by the GRS, with which it is connected with an “and” in that sentence. However, in lines 49, 124 and 137 the term clearly includes the additive component. “Epistasis” alone, without the “marginal”, is used sometimes to refer to the sum of all effects (lines 117, 139, 450, 470) and sometimes to specifically the additional contribution of the $\sigma_{C \times G}$ component (lines 75, 263, 452, Fig. 1d). The paper should use two different terms consistently, to refer to these two notions.

2. FAME is a highlight of the paper that needs closer scrutiny. First, the method has not cleared peer review. The paper validating it is under review, and I do not think that it is wise to publish the current submission before that paper is accepted.

3. Beyond the question of FAME’s validity, we must also understand the importance, in quantitative terms, of its contribution to defining risk. Dealing with the computational bottleneck of studying interactions is touted as FAME’s highlight. Therefore, its results must be compared to what we get if we disregard interactions, under a simple additive model from the GRS. To use the terminology of Fig. 1d, does Table 3 compare $C \cdot \beta_C$ to $\Sigma G \cdot \beta_G + \Sigma G \cdot C \cdot \beta_{C \times G}$ or to $\Sigma G \cdot C \cdot \beta_{C \times G}$? I am assuming it is the former but, either way, the EIP metric tells us nothing about how much is added by considering non-additive interactions, over just using the GRS. In an earlier version of this manuscript that I reviewed for another journal and to which I no longer

have access, I recall linear regression figures of quantitative phenotypes against GRS, removed from this version. The portion of variance explained in these analyses (quantified as r^2) was suggestive of the EIP values shown in Table 3, raising the question of how much margin there was to improve by adding the interactions. Without the direct comparison I recommend, the results in Table 3 cannot be used to judge the importance of FAME analysis.

4. Line 477. The EIP is calculated as the percentage increase in the risk information, the 100% being what comes from the monogenic variant only. In that case, the number in line 477 should be "2.7 times more accurate", not 1.7. This huge difference is, of course, due to the small denominator (minuscule effect size of the monogenic variant, at least as determined by the ESM1b score).

5. It is not clear how ESM1B can tell GOF from LOF variants, but the only evidence presented for this is unconvincing, for lack of a more tactful term. The carriers of the pink points in Fig.3A and 3B have a BMI that is in the overweight (UKB) or frankly obese (BioMe) range. What makes these variants protective from obesity? These are not clinical subjects selected to genotype for being obese in the first place; both databases are samplings of the general population.

Suggestions for improvement

6. The plots in Fig 2B will be much more informative if a "bottle" plot of non-carriers is added for comparison (no need for the individual points, of course unless you wish to add extreme outliers).

7. Table 2. Please correct the erroneous phenotype identifier at the top left corner of the table. Also, the column heading "Total unique & curated pathogenic variants" is ambiguous. Is it the number of unique curated variants? Then the "&" is unnecessary. However, the numbers in that column are much higher than the count of the dark blue points in Fig. 2. Please resolve this apparent contradiction.

8. The sentence in lines 81-84 is incomprehensible and gives the unjustified impression that epistasis can tell us anything about biochemical mechanism. A look at Ref. 9 makes it clear that this is a rare case of the epistasis due to a single genetic variant in trans, i.e. a clearly digenic effect. The sentence can be drastically shortened and made comprehensible by the simple mention of the term "digenic". By the way, it is likely that digenic effects were what William Baetson was thinking when he coined the term "epistasis" over a century ago.

Reviewer #3

(Remarks to the Author)

This paper seeks to explore reasons why people carrying apparently strongly penetrant rare "monogenic" variants have varying disease phenotypes

The paper is clear and thorough.

The authors have three hypotheses about how there may be varying levels of disease in people who are carriers of rare variants in genes

My comments are split by these three potential mechanisms and I hope are useful:

The first is that variants that are more damaging to the protein will have stronger effects on disease. The authors show neatly how their variant effect predictor compares very favourably with other variant effect predictors. This result seems like a solid proof of principle analyses but does not tell us anything particularly new - it would have been presumably rather surprising if there was no association between predicted effect and penetrance after the huge amount of work put into these predictive algorithms

The second is that the polygenic component to the same trait will influence disease penetrance and severity. This again makes sense - a high polygenic risk will increase risk of a rare variant carrier making it to a diagnosis, and has been shown in several diseases (notably breast cancer BOADICEA: a comprehensive breast cancer risk prediction model incorporating genetic and nongenetic risk factors | Genetics in Medicine (nature.com) but also familial hyperlipidaemia and developmental delay Genetic modifiers of rare variants in monogenic developmental disorder loci - PubMed (nih.gov))

The third is that genetic background - effectively a measure of background population and distant relatedness - could influence penetrance. There is evidence of an interaction but to what extent does the variance in the phenotype between the background "populations" influence this measure? It is possible that different background populations have different distributions including different variances. Because the disease phenotypes are defined using thresholds of quantitative traits the distribution of the trait could make a subtle difference to the effects of rare variants - but across multiple variants could become statistically significant but reflect artefactual effects. One way of testing this is to repeat the analysis using inverse normalised versions of the continuous traits within background "populations" - where "populations" are defined by genetic distance. There is a good discussion of these "scale" issues here Adiposity amplifies the genetic risk of fatty liver disease conferred by multiple loci | Nature Genetics, here Gene-obesogenic environment interactions in the UK Biobank study - PubMed (nih.gov) (with the use of negative control variables).

Minor

1. Why convert measures to non international standards? Mmol for lipids and hba1c is well known now and more widely applied and understood.

2. Have is a better verb than be for describing people with diabetes - eg had diabetes better than were diabetic

3. Approximately 25% of the Ukb have BMIs > 30. Is this an appropriate definition of monogenic obesity?

Reviewer #4

(Remarks to the Author)

As I was not one of the initial reviewers, I have confined my review to consider whether the authors have addressed the previous reviewers' concerns, rather than making additional observations and comments at this stage in the process.

I believe that the authors have done a good job of addressing the majority of the reviewers' comments and concerns. There remain a few outstanding issues:

1) Given the answers to some reviewers' questions about the BioMe cohort, it appears they were not included in the PRS or marginal epistasis analyses and the text of the results section suggests they were not included in the penetrance calculations either. I'm a little confused as to what benefit this group added to the analysis. Were they just used for ESM1b prediction replication? If so, what value did this cohort add over the newly added replication in an additional 300,000 participants? Is the value simply the more diverse nature of BioMe participants?

2) No additional variant QC appears to have been done other than the standard UKBB protocol. No explanation is given on whether allele balance, coverage, depth or other metrics were used to further assess variant quality. There is also no mention of whether IGVs were assessed for each variant. Inclusion of any poor quality variants has the potential to significantly impact on penetrance estimates as well as estimates of where extremes of PRS overlap variant carrier phenotype. This is particularly important here as the authors allude to gene and variant specific penetrance. I'm also unsure about the claim made in the final summary about this study representing real-world estimates of penetrance. UKBB is not a representative cohort and is known to be healthier and wealthier than the UK population. This coupled with concerns over variant quality would lead to me think this should be toned down to something like "provide evidence of reduced penetrance in a population cohort".

3) The replication of 5/6 results in the additional 300,000 participants in UKBB was a useful addition. The methods need updating to reflect this as the cohort information still refers only to the first 200,000. However, I wonder whether the authors would like to comment further on why they don't believe MCR4 replicated in a very similar cohort with UKBB that was 50% larger than their discovery cohort. This is made even more confusing as this finding is reported to replicate in BioMe despite a much smaller and more diverse cohort. I think this needs discussing.

4) I don't think reviewer 3's concerns about medications other than statins or dosage effects has been answered sufficiently.

5) Figure 2 and results. It feels very odd to use the term penetrance when discussing non-carriers. Penetrance relates to the chance of a variant resulting in a phenotype so what is being discussed in non-carriers is simply the background prevalence of the phenotype.

6) Figure 4 last sentence of legend "Clinvar variants not included in (D)" is not true. Presumably this should be "in (E) and (F)"

7) I'm unsure why the clinvar weak group is being used. It seems to me that no-one would sensibly consider a variant pathogenic if it were now designated clinvar benign but had previously been asserted as pathogenic. As the method reads, these variants are included here. Having only one pathogenic assertion is a very low bar as evidenced by the findings and I'm not sure what it adds.

Version 1:

Reviewer comments:

Reviewer #1

(Remarks to the Author)

The revised version has added the word "marginal" in front of every instance of "epistasis", without making the paper more readable. A clear and rigorous definition of the term "marginal epistasis" should be provided on first mention. It is decidedly not a standard term and will not be understood and taken for granted by the average reader of Nature Communications. Perhaps it would help if the term "modify" (modifier, modification, etc) is used for simply additive effects in trans, without knowing and considering Gene x Gene interactions. Then "marginal" can be defined (strictly in the confines of this paper) as including the additional information obtained from adding GxG. The two are clearly presented as distinct in lines 47-49 of the abstract (connected with the word "and").

This brings me to the main problem with the paper, which was completely ignored in writing the revised version. How much more meaningful are the EIP values shown in Table 3 compared to calculating only trans modifier effects without knowledge of non-linear GxG interactions? What makes the paper potentially groundbreaking would be the use of FAME to enable detection of GxG, if that component made a major difference. I conjecture that the difference would be small. I will be happy

to be proven wrong, by simply presenting the same calculations using only the linear effects of the GRS variants without knowledge of any non-linear interactions. Without such comparison, the reader is left wondering.

Also, please cite and discuss PMID 39379762, a very recent paper, very relevant to the concept, and covering one of the phenotypes studied here.

Reviewer #4

(Remarks to the Author)

The authors have taken on the majority of concerns raised and this has resulted in a substantially improved manuscript which presents their work in an easier to understand manner.

I have two small suggestions remaining;

1) I remain concerned about the lack of additional variant QC and visualisation based on similar work we have done showing a not insignificant number of called variants to have little evidence supporting them. I still believe this could impact on the results presented here as only a few errors in such small sub-groups could radically change some findings. Whilst I appreciate the CRAM files are now stored on DNA Nexus, they are obtainable on application. I understand the authors may feel it out of scope (or resource) to undertake this analysis but in the absence of this I think it warrants more specific discussion as a limitation.

2) I think the language around some of the low LDL variants, specifically PCSK9 variants could be adapted slightly for clarity. There is mention of pathogenic variants for example yet even the curated subset contain a number of variants rated benign or VUS on Clinvar, albeit it often for high LDL. However the cut off of 80mg/l itself possibly compounds this confusion. The PCSK9 variants may well cause lower LDL but is this pathogenic to the level of monogenic disease, especially when levels between 70-100 are often quoted in ideal ranges. Consider rewording to LDL-lowering variants or similar to avoid confusion. This is demonstrated by the low penetrance shown for PCSK9 although as before penetrance here seems to refer to lower than a pretty conservative cut-off.

Overall though I think these issues are minor and am satisfied my questions have been answered

Version 2:

Reviewer comments:

Reviewer #1

(Remarks to the Author)

1. The revised version gives a much-needed and satisfactory definition of "marginal epistasis", a decidedly non-standard term. To call it a mathematical definition, the legend of Fig. 1 must define what quantity is represented by Y.

2. I don't understand what is meant by "our focus is on determining the biological underpinning of heterogeneity amongst carriers". The only outcome of FAME is table 3, which compares the proportion of variance explained. I see no biology.

3. In the Results section (lines 118-122) methodology is described to determine the additive polygenic background from GWAS studies, but no results are shown or referred to a table or figure. I understand that this cannot be done with FAME but can't the ratio of variance explained by the linear effects alone, without taking interactions into account, be calculated with much simpler arithmetic?

We thank the reviewers for their careful review of our manuscript and have updated key aspects of our manuscript in this revision below. We believe that the clarity paper is much improved and that the results are bolstered by additional requested analyses. We hope the reviewers find the updates improves the clarity of our findings and highlights the exciting findings of our analysis.

REVIEWER COMMENTS

Reviewer #1 (Remarks to the Author):

This paper leverages the powerful datasets of the UKBb and the BioMe to examine the genetic basis of variable penetrance of monogenic effects on five cardiometabolic quantitative traits, triglycerides, LDL, HDL, BMI and HbA1c (MODY variants). First, the effect magnitude of the specific monogenic variant is quantified by ESM1b, an interesting novel method of pathogenicity prediction based on protein language that the authors have recently validated (Ref 14). ESM1B scores explain from 3.6% (APOA5 and LPL for triglycerides) to 24% (LDLR for LDL) of the quantitative trait variance (based on the r^2 values from Fig. 4), leaving a lot of room for polygenic effects in trans. These are first addressed by examining the effect of polygenic risk scores on the phenotype by showing, unsurprisingly, that complex-trait GRS values in non-carriers explain as much or more of the extreme phenotypic variance in these quantitative traits. The question of their relative contributions is addressed by FAME, a novel computational approach that calculates the EIP, which is shown, for four of the traits, to add very substantially to the prediction based solely on the monogenic variant.

This study confirms and systematizes, with repository-sized samples, previous work on trans effects on the penetrance or (more correctly for quantitative traits) expressivity of monogenic variants and is of potential clinical utility in determining pathogenicity of monogenic variants. It is a substantial contribution to the field. However, I find that, as explained below, the paper implicitly makes an important but unsupported claim. Also the paper will be much easier to read and understand, if the term “epistasis” with or without the qualifiers “marginal” and “genetic” is used in a rigorously defined and consistent matter.

Major concerns

1. Semantics: The term “epistasis” has been used by different authors to mean different things (reviewed by Phillips Nat Rev Genet. 2008 Nov; 9(11): 855–867, PMID: 18852697, doi 10.1038/nrg2452). By itself, it only means “genetic modifier(s) in trans”, and any other use should be rigorously defined or (better in my opinion) completely avoided and replaced by terms more meaningful in 21st-century genetics. The paper attempts (line 270) to define “marginal epistasis” as the $\sigma_{C \times G}$ component, i.e. clearly the effect due to non-additive interaction of complex-trait loci with the “target feature” (a jargon term for “monogenic variant”, I presume) at the express exclusion of σ_G , i.e. the simply additive effect of known GWAS loci as quantified by the GRS, with which it is connected with an “and” in that sentence. However, in lines 49, 124 and 137 the term clearly includes the additive component. “Epistasis” alone, without the “marginal”, is used sometimes to refer to the sum of all effects (lines 117, 139, 450, 470) and sometimes to specifically the additional contribution of the $\sigma_{C \times G}$ component (lines 75, 263, 452, Fig. 1d). The paper should use two different terms consistently, to refer to these two notions.

We have updated the main text to consistently use “marginal epistasis” which is our focus here and is the $\sigma_{C \times G}$ component of the FAME model. It is the variance of the trait explained by non-additive interactions of all trans variants and the feature of interest, which is monogenic variants in this case. We are not 100% sure what the reviewer means by “complex-trait loci”, but any genetic variant in trans that modifies the effect size of the target variant will be included in the estimate. For clarity σ_G does not quantify the GRS (or PRS) but rather determines its upper bound of predictive accuracy.

2. FAME is a highlight of the paper that needs closer scrutiny. First, the method has not cleared peer review. The paper validating it is under review, and I do not think that it is wise to publish the current submission before that paper is accepted.

FAME is currently undergoing second round revisions at Nature Genetics and the most recent round of revisions were submitted in October 2024. We are confident it will be accepted shortly and are happy to delay publication of this paper until that one is published. It is available on biorxiv.

3. Beyond the question of FAME's validity, we must also understand the importance, in quantitative terms, of its contribution to defining risk. Dealing with the computational bottleneck of studying interactions is touted as FAME's highlight. Therefore, its results must be compared to what we get if we disregard interactions, under a simple additive model from the GRS. To use the terminology of Fig. 1d, does Table 3 compare $C \cdot \beta_C$ to $\Sigma G \cdot \beta_G + \Sigma G \cdot C \cdot \beta_{CxG}$ or to $\Sigma G \cdot C \cdot \beta_{CxG}$? I am assuming it is the former but, either way, the EIP metric tells us nothing about how much is added by considering non-additive interactions, over just using the GRS. In an earlier version of this manuscript that I reviewed for another journal and to which I no longer have access, I recall linear regression figures of quantitative phenotypes against GRS, removed from this version. The portion of variance explained in these analyses (quantified as r^2) was suggestive of the EIP values shown in Table 3, raising the question of how much margin there was to improve by adding the interactions. Without the direct comparison I recommend, the results in Table 3 cannot be used to judge the importance of FAME analysis.

We believe there is a lot of confusion, possibly due to our previous presentation style, about several quantities and figures here. The figure the reviewer is referring to with GRS/PRS was moved and is now Supplemental Figure 4. This figure is a plot of PRS against phenotype amongst carriers of corresponding monogenic variants. It demonstrates the PRS (GRS) is predictive of trait even amongst carriers and is therefore a source of phenotypic heterogeneity in variant carriers. While widely believed to be true, it had not been previously demonstrated. However, the reviewers in the last round felt this result was not novel and so we moved it to the supplementary material. It is only very peripherally related to our analysis of epistasis and is not determined by EIP or σ_{CxG} . Rather, it is determined by σ_G .

There is also confusion about what EIP is. EIP is defined as $100 \cdot \sigma_{CxG}^2 / \beta_C^2$, and relates the phenotypic variance determined by epistatic interaction to that determined by carrier status. If EIP is 0% then there is no marginal epistasis, if EIP is 100%, then marginal epistasis explains as much variance as carrier status itself. The reviewer is clearly interested in the impact this will have on predictive performance relative to an additive model and writes, "its results must be compared to what we get if we disregard interactions, under a simple additive model from the GRS". We respectfully disagree with this sentiment. First and foremost because the ability of a linear model to approximate a non-additive genetic architecture is determined by the nature of the non-additive component. Some are easily and accurately approximated with linear functions and others are not. Here, we only know the total variance explained through epistasis, not its full specification. Second, and more importantly, our objective was to determine the extent to which genetic background modify the impact of monogenic variants. We believe this is an important scientific question outside of practical considerations of predictive models. However, the evidence we report suggests that going forward those traits with high EIP could have improved predictors with either non-additive models or linear approximations of those models, and that is exciting.

4. Line 477. The EIP is calculated as the percentage increase in the risk information, the 100% being what comes from the monogenic variant only. In that case, the number in line 477 should be "2.7 times more accurate", not 1.7. This huge difference is, of course, due to the small denominator (minuscule effect size of the monogenic variant, at least as determined by the ESM1b score).

We agree that line 477 should be “2.7” instead of “1.7”, and we have updated the main text to reflect this change. The ESM1b scores were not included in the FAME analysis portion; the EIP ratio is defined as variance in phenotype due the genetic background interacting with the carrier status over the variance in phenotype due to carrier status.

5. It is not clear how ESM1B can tell GOF from LOF variants, but the only evidence presented for this is unconvincing, for lack of a more tactful term. The carriers of the pink points in Fig.3A and 3B have a BMI that is in the overweight (UKB) or frankly obese (BioMe) range. What makes these variants protective from obesity? These are not clinical subjects selected to genotype for being obese in the first place; both databases are samplings of the general population.

These variants have been proven to be gain-of-function in 10.1016/j.cell.2019.03.044 using assays that measured the activity of the MC4R protein when different variants were introduced. These authors not only found a significant gain in MC4R activity via assays, but found carriers of these gain-of-function variants had lower BMIs compared to other individuals. These individuals represent real world effects, not simply in vitro activity assays, which correlate with the gain-of and loss-of function hypothesis. We also show that in *PCSK9* that ESM1b has positive correlation with mean LDL, which reflects how gain-of-function *PCSK9* variants cause high LDL while loss-of-function variants cause low LDL (10.1038/s41569-018-0107-8).

Suggestions for improvement

6. The plots in Fig 2B will be much more informative if a “bottle” plot of non-carriers is added for comparison (no need for the individual points, of course unless you wish to add extreme outliers).

The non-carrier phenotypes have been added to Figure 2B.

7. Table 2. Please correct the erroneous phenotype identifier at the top left corner of the table. Also, the column heading “Total unique & curated pathogenic variants” is ambiguous. Is it the number of unique curated variants? Then the “&” is unnecessary. However, the numbers in that column are much higher than the count of the dark blue points in Fig. 2. Please resolve this apparent contradiction.

This table is inclusive of all pathogenic variant effects, including premature stop-gained, frameshift, splicing, etc. The figure with the blue dots is restricted to only missense variants. The difference in which pathogenic variant effects are considered in the table and the figure account for the contradiction.

The formal name for Low LDL was corrected in Table 2. We also made the heading for the total number of curated, pathogenic variants more clear.

8. The sentence in lines 81-84 is incomprehensible and gives the unjustified impression that epistasis can tell us anything about biochemical mechanism. A look at Ref. 9 makes it clear that this is a rare case of the epistasis due to a single genetic variant in trans, i.e. a clearly digenic effect. The sentence can be drastically shortened and made comprehensible by the simple mention of the term “digenic”. By the way, it is likely that digenic effects were what William Baetson was thinking when he coined the term “epistasis” over a century ago.

Thank you for this comment and that’s an interesting observation about William Baetson. For clarity, this sentence has been shortened and includes use of “digenic”.

Reviewer #3 (Remarks to the Author):

This paper seeks to explore reasons why people carrying apparently strongly penetrant rare “monogenic” variants have varying disease phenotypes. The paper is clear and thorough. The authors have three hypotheses about how there may be varying levels of disease in people who are carriers of rare variants in genes

My comments are split by these three potential mechanisms and I hope are useful:

The first is that variants that are more damaging to the protein will have stronger effects on disease. The authors show neatly how their variant effect predictor compares very favourably with other variant effect predictors. This result seems like a solid proof of principle analyses but does not tell us anything particularly new - it would have been presumably rather surprising if there was no association between predicted effect and penetrance after the huge amount of work put into these predictive algorithms

This result was not anticipated at all when protein language models were generated; the scores' ability to predict monogenic missense variant effect size, especially differentiating between gain- and loss-of-function effects, is a novel result.

The second is that the polygenic component to the same trait will influence disease penetrance and severity. This again makes sense - a high polygenic risk will increase risk of a rare variant carrier making it to a diagnosis, and has been shown in several diseases (notably breast cancer BOADICEA: a comprehensive breast cancer risk prediction model incorporating genetic and nongenetic risk factors | Genetics in Medicine (nature.com) but also familial hyperlipidaemia and developmental delay Genetic modifiers of rare variants in monogenic developmental disorder loci - PubMed (nih.gov))

We agree that the polygenic risk score analysis of this paper was not as novel as the remaining parts. However, our use of finer-sized PRS percentile bins in Figure 4 adds evidence that PRS is closer to clinical implementation.

The third is that genetic background - effectively a measure of background population and distant relatedness - could influence penetrance. There is evidence of an interaction but to what extent does the variance in the phenotype between the background “populations” influence this measure? It is possible that different background populations have different distributions including different variances. Because the disease phenotypes are defined using thresholds of quantitative traits the distribution of the trait could make a subtle difference to the effects of rare variants - but across multiple variants could become statistically significant but reflect artefactual effects. One way of testing this is to repeat the analysis using inverse normalised versions of the continuous traits within background “populations” - where “populations” are defined by genetic distance. There is a good discussion of these “scale” issues here Adiposity amplifies the genetic risk of fatty liver disease conferred by multiple loci | Nature Genetics, here Gene-obesogenic environment interactions in the UK Biobank study - PubMed (nih.gov) (with the use of negative control variables).

While this is an interesting question, exploring how fine-scale population structure can cause bias in FAME is beyond the scope of this paper. This question is going to be explored in future papers.

Minor

1. Why convert measures to non international standards? Mmol for lipids and hba1c is well known now and more widely applied and understood.

We utilized the non-international standards to be consistent with the units used for the polygenic risk score weight units cited.

2. Have is a better verb than be for describing people with diabetes - eg had diabetes better than were diabetic
We have removed “diabetic” and replaced it with “had diabetes” in the main text.

3. Approximately 25% of the Ukb have BMIs > 30. Is this an appropriate definition of monogenic obesity ?
This is a good point and the World Health Organization has defined obesity as a BMI > 30, however, obesity itself is not necessarily a disease. Monogenic obesity differentiates from general obesity in that monogenic obesity often has very early onset and affected individuals have BMIs that are extremely obese (10.1038/s41576-021-00414-z). Additionally, these individuals may not lose weight like others via lifestyle changes and require more intensive treatment, such as medication and surgery, to manage their weight (10.1159/000445061). Therefore, our definition is based on the presence of a BMI >30 and the presence of a pathogenic variant in an established gene.

Reviewer #4 (Remarks to the Author):

As I was not one of the initial reviewers, I have confined my review to consider whether the authors have addressed the previous reviewers’ concerns, rather than making additional observations and comments at this stage in the process.

I believe that the authors have done a good job of addressing the majority of the reviewers’ comments and concerns. There remain a few outstanding issues:

1) Given the answers to some reviewers’ questions about the BioMe cohort, it appears they were not included in the PRS or marginal epistasis analyses and the text of the results section suggests they were not included in the penetrance calculations either. I’m a little confused as to what benefit this group added to the analysis. Were they just used for ESM1b prediction replication? If so, what value did this cohort add over the newly added replication in an additional 300,000 participants? Is the value simply the more diverse nature of BioMe participants?

Yes, the BioMe cohort was included in the ESM1b replication only, and the results include triglycerides correlations that were significant. We sought a separate validation in an independently ascertained and diverse biobank to determine whether our findings were specific to the nature of the UKBB or the specific population included in their cohort. The BioMe participants were not included in the PRS and FAME analyses because this cohort was too small, underpowered and mostly made up of non-Europeans (10.1016/j.cell.2021.03.034).

2) No additional variant QC appears to have been done other than the standard UKBB protocol. No explanation is given on whether allele balance, coverage, depth or other metrics were used to further assess variant quality. There is also no mention of whether IGVs were assessed for each variant. Inclusion of any poor quality variants has the potential to significantly impact on penetrance estimates as well as estimates of where extremes of PRS overlap variant carrier phenotype. This is particularly important here as the authors allude to gene and variant specific penetrance. I’m also unsure about the claim made in the final summary about this study representing real-world estimates of penetrance. UKBB is not a representative cohort and is known to be healthier and wealthier than the UK population. This coupled with concerns over variant quality would lead to me think this should be toned down to something like “provide evidence of reduced penetrance in a population cohort”.

Unfortunately, access to CRAM files for UKBB is now limited, and we do not have permission to view them to verify the variant calls. We cannot verify calls via IGV due to these limitations. We have also updated the sentence in the final summary to reflect the limitations of the variant call checking.

3) The replication of 5/6 results in the additional 300,000 participants in UKBB was a useful addition. The methods need updating to reflect this as the cohort information still refers only to the first 200,000. However, I wonder whether the authors would like to comment further on why they don't believe MCR4 replicated in a very similar cohort with UKBB that was 50% larger than their discovery cohort. This is made even more confusing as this finding is reported to replicate in BioMe despite a much smaller and more diverse cohort. I think this needs discussing.

Obesity is a trait that is largely influenced by environmental effects, ex. dietary choices and exercise levels. We did not adjust for these factors, which adds noise to the analysis. Additionally, a large portion of individuals of the UK Biobank are already overweight (BMI \geq 25kg/m², 66.3%) or obese (BMI \geq 30kg/m², 23.9%), reflecting how this phenotype is heavily impacted by factors outside of rare genetic variants.

4) I don't think reviewer 3's concerns about medications other than statins or dosage effects has been answered sufficiently.

The coefficients we used to adjust for statin-usage were calculated based on meta-analysis of randomized controlled trials of more than 170,000 individuals. This method of adjusting for statins was also used in several other published studies: [10.1038/s41467-021-23556-4](https://doi.org/10.1038/s41467-021-23556-4), [10.1001/jamanetworkopen.2020.3959](https://doi.org/10.1001/jamanetworkopen.2020.3959), [10.1093/eurheartj/ehw028](https://doi.org/10.1093/eurheartj/ehw028). We have cited the appropriate references and this information is included in our Methods.

5) Figure 2 and results. It feels very odd to use the term penetrance when discussing non-carriers. Penetrance relates to the chance of a variant resulting in a phenotype so what is being discussed in non-carriers is simply the background prevalence of the phenotype.

We agree and have restricted the use of the term penetrance only when referring to the prevalence of disease in individuals who are carriers of pathogenic variants for that trait. We have updated Figure 2 caption to remove "penetrance" when discussed in context of the non-carriers, and replaced it with "disease prevalence".

6) Figure 4 last sentence of legend "Clinvar variants not included in (D)" is not true. Presumably this should be "in (E) and (F)"

This mistake was corrected and the Figure 4 caption was updated to "ClinVar variants are not included in (E) and (F)".

7) I'm unsure why the clinvar weak group is being used. It seems to me that no-one would sensibly consider a variant pathogenic if it were now designated clinvar benign but had previously been asserted as pathogenic. As the method reads, these variants are included here. Having only one pathogenic assertion is a very low bar as evidenced by the findings and I'm not sure what it adds.

Many of the studies that supply ClinVar findings are restricted to small family studies, and which may lead to a pathogenic variant with low- to moderate-penetrance labeled with conflicting assertions. Additionally, ClinVar is one of the most used databases to identify pathogenic/likely pathogenic variants, but it does have limitations as there is no oversight and regulations for submitted variants. We include these types of variants to show that ClinVar has room for improvement in terms of more accurately classifying variants, especially those of unknown significance.

REVIEWER COMMENTS

Reviewer #1 (Remarks to the Author):

The revised version has added the word “marginal” in front of every instance of “epistasis”, without making the paper more readable. A clear and rigorous definition of the term “marginal epistasis” should be provided on first mention. It is decidedly not a standard term and will not be understood and taken for granted by the average reader of Nature Communications. Perhaps it would help if the term “modify” (modifier, modification, etc) is used for simply additive effects in trans, without knowing and considering Gene x Gene interactions. Then “marginal” can be defined (strictly in the confines of this paper) as including the additional information obtained from adding GxG. The two are clearly presented as distinct in lines 47-49 of the abstract (connected with the word “and”).

We disagree that all we did was add the “marginal” in front of every instance of epistasis. In fact, our first mention of marginal epistasis cited Figure 1, which provided a precise mathematical definition. We now additionally include the following definition in the text, “the combined pairwise interaction effects between carrier status and all other SNPs while controlling for linear, additive effects.” in lines 74-75.

We have discussed the addition of the term “modify” with our colleagues and decided against using it. They all felt that adding the term “modify” will make readers think of interactive effects, not additive effects.

Additive and interaction effects are distinct as we presented in lines 47-49 of the abstract. We have now expanded the definition for clarity in the abstract as well.

This brings me to the main problem with the paper, which was completely ignored in writing the revised version. How much more meaningful are the EIP values shown in Table 3 compared to calculating only trans modifier effects without knowledge of non-linear GxG interactions? What makes the paper potentially groundbreaking would be the use of FAME to enable detection of GxG, if that component made a major difference. I conjecture that the difference would be small. I will be happy to be proven wrong, by simply presenting the same calculations using only the linear effects of the GRS variants without knowledge of any non-linear interactions. Without such comparison, the reader is left wondering.

We disagree again that we ignored this comment, we tried to explain in our response why we focus on these analyses (additive and interactive) separately and will try again now.

First and foremost, the quantities that the reviewer is asking for are not estimable with the parameters returned from FAME. FAME provides variances effect estimates with respect to controls, and does not compute a within carrier estimate. While we agree that this calculation would be interesting and exciting, unfortunately, the FAME framework is unable to estimate the trans-modifier effect this reviewer is requesting.

Second, and more importantly, our focus is on determining the biological underpinning of heterogeneity amongst carriers, not developing the most powerful approaches to predict phenotype. It is likely the case that PRS will eventually be more predictive of phenotype than carrier status. Indeed, we are already seeing that individuals in the top percentiles of PRS for several traits (coronary artery disease, atrial fibrillation, type 2 diabetes, inflammatory bowel disease, and breast cancer) are at 3-fold risk of disease, a risk that is comparable to monogenic variant carriers (10.1038/s41588-018-0183-z). Concomitantly, it is also very likely (as the reviewer points out) that additive effects across the entire genome will be much stronger predictors of phenotype than the GxCarrier status based predictor. However, and central to the significance of this paper, the observation of marginal epistasis on the scale we observe, implies that other variants in the genome can serve to silence and or enhance the impact of these clinically used variants. While PRS are important tools, understanding genetic architecture is also a fundamental research pursuit in human genetics (see <https://grants.nih.gov/grants/guide/pa-files/PAR-25-255.html> for example).

Also, please cite and discuss PMID 39379762, a very recent paper, very relevant to the concept, and covering one of the phenotypes studied here.

We have reviewed this paper and added this citation to the introduction on line 81.

Reviewer #4 (Remarks to the Author):

The authors have taken on the majority of concerns raised and this has resulted in a substantially improved manuscript which presents their work in an easier to understand manner.

I have two small suggestions remaining;

1) I remain concerned about the lack of additional variant QC and visualisation based on similar work we have done showing a not insignificant number of called variants to have little evidence supporting them. I still believe this could impact on the results presented here as only a few errors in such small sub-groups could radically change some findings. Whilst I appreciate the CRAM files are now stored on DNA Nexus, they are obtainable on application. I understand the authors may feel it out of scope (or resource) to undertake this analysis but in the absence of this I think it warrants more specific discussion as a limitation.

We now state very clearly in discussion the above as a limitation of our study. This is on lines 519-522, and it now reads "Our measures of penetrance may also be affected by our variant calling process. Because we did not verify variant calls via inspection of CRAM files, some variants of interest may be false positives."

2) I think the language around some of the low LDL variants, specifically PCSK9 variants could be adapted slightly for clarity. There is mention of pathogenic variants for example yet even the curated subset contain a number of variants rated benign or VUS on Clinvar, albeit it often for high LDL. However the cut off of 80mg/l itself possibly compounds this confusion. The PCSK9 variants may well cause lower LDL but is this pathogenic to the level of monogenic disease, especially when levels between 70-100 are often quoted in ideal ranges. Consider rewording to

LDL-lowering variants or similar to avoid confusion. This is demonstrated by the low penetrance shown for PCSK9 although as before penetrance here seems to refer to lower than a pretty conservative cut-off.

We agree. We have now updated the language in the main text, the main figures, and the supplemental tables to refer to what was in the previous draft as “low LDL pathogenic variants” to LDL-lowering variants. We also add a sentence to the introduction to reflect that our work not only covers disease-causing variants, but also variants that have a beneficial effect in lines 99-100.

Overall though I think these issues are minor and am satisfied my questions have been answered

We thank the reviewer for their comments and hope we have addressed any remaining comments to their satisfaction here.

Reviewer #1 (Remarks to the Author)

1. The revised version gives a much-needed and satisfactory definition of “marginal epistasis”, a decidedly non-standard term. To call it a mathematical definition, the legend of Fig. 1 must define what quantity is represented by Y.

Thank you for pointing this out, we have defined all variables in the caption and the main text.

2. I don't understand what is meant by “our focus is on determining the biological underpinning of heterogeneity amongst carriers”. The only outcome of FAME is table 3, which compares the proportion of variance explained. I see no biology.

Sorry for the confusion, we clarify it here. We have learned from our analysis answers to questions about the nature of human genetic architecture and its relationship to complex traits. Specifically, we did not know before this paper, the extent to which the impact of these monogenic variants were epistatically modified by other variants in the genome. The results presented in Table 3 show that this epistatic effect is very large, often exceeding the effect size of these variants themselves. We believe that this is a question of biology, but are happy to assign it to a different field if there is a more appropriate one.

To further clarify how this can impact our understanding of genetics we include the following example from a similar analysis on a different gene in the FAME paper and have updated the main text introduction:

“Our analysis revealed significant marginal effects (ME) of rs738409 on ALT, with a proportion of variance explained comparable to that of rs3827385 ($p = 4.90 \times 10^{-24}$, $\sigma^2_{\text{gxg,t}} = 1.48 \times 10^{-3}$). Notably, prior work [62] has identified an interaction between rs738409 and a splice variant rs72613567 in HSD17B13 on chromosome 4, affecting liver function as measured by ALT and aspartate aminotransferase (AST) levels. Specifically, it has been found that the HSD17B13 rs72613567:TA allele, associated with reduced aminotransferase levels, attenuates the effect of rs738409 on aminotransferase levels and is linked to reduced PNPLA3 mRNA expression. We replicated this in the UK Biobank”.

62. Noura S Abul-Husn, Xiping Cheng, Alexander H Li, Yurong Xin, Claudia Schurmann, Panayiotis Stevis, Yashu Liu, Julia Kozlitina, Stefan Stender, G Craig Wood, et al. A protein-truncating hsd17b13 variant and protection from chronic liver disease. **New England Journal of Medicine** <https://www.nejm.org/doi/full/10.1056/NEJMoa1712191>

3. In the Results section (lines 118-122) methodology is described to determine the additive polygenic background from GWAS studies, but **no results are shown** or referred to a table or figure. I understand that this cannot be done with FAME but can't the ratio of variance explained by the linear effects alone, without taking interactions into account, be calculated with much simpler arithmetic?

Lines 118-122 are in the Introduction and not the Results section. In the figure, we do show that additive polygenic background from GWAS studies exists. This is commonly referred to as chip heritability when its ratio of variance is computed with respect to the overall phenotypic variance. It was not the focus of this paper, but we can compute it. We have added this to Supp Data 7.

We note that we are only 75% sure this is what the reviewer is asking about because of the line number issue. If we were wrong, it would help if they can write down precisely (mathematically) what they would like. If it is estimable we can add that as well.